# Congestion in multi-function parallel network DEA

**Sarvar Sadat Kassaei**[1], **Farhad Hosseinzadeh Lotfi**[1]*, **Alireza Amirteimoori**[2],
**Mohsen Rostamy-Malkhalifeh**[1], **Bijan Rahmani Parchikolaei**[3]

**1** Department of Mathematics, Science and Research Branch, Islamic Azad University, Tehran, Iran,
**2** Department of Applied Mathematics, Islamic Azad University, Rasht, Iran, **3** Department of Mathematics,
Central Tehran Branch, Islamic Azad University, Tehran, Iran

\* farhad@hosseinzadeh.ir

**Data Availability Statement:** All relevant data are within the paper.

**Funding:** The author(s) received no specific funding for this work.

## Abstract

Congestion is an economic phenomenon of the production process in which the excessive values of inputs lead to a reduction of the outputs. As the existence of congestion makes to increase costs and decreases efficiency, this issue is not acceptable for decision makers. Hence, many methods have been proposed to detect the congestion in the Data Envelopment Analysis framework (DEA). Most of these methods are designed to deal with the decision making units (DMUs) that have no network structure. However, in most real-world applications, some units are composed of independent production subunits. Therefore, a new scheme is required to determine the congestion of such units. A multi-function parallel system is a more common case in the real world that is composed of the same number of subunits such that each subunit has specific functions. In this paper, considering the operation of individual components of each DMU, a new DEA model is proposed to identify and evaluate the congestion of the multi-function parallel systems. It is shown that the proposed method is highly economical in comparison with the existing black-box view from a computational viewpoint. Then, the proposed model is illustrated using a numerical example along with a real case study.

## 1 Introduction

Stochastic frontier analysis (SFA) is utilized to analyze the technical inefficiency in the framework of production functions. Production units produce according to a common technology and reach the frontier when the maximum possible output for a given set of inputs is produced. The main advantages of the SFA model are its capacity to accommodate statistical noise, such as measurement error, and its parametric specification of the technology, allowing standard statistical tests to be used [1]. Zúniga-González et al., [2] proposed a stochastic frontier model of environmental inefficiency effects for dairy farms in Mexico. In a similar direction, Zúniga-González et al., [3] showed there is concern that in the coming decades, temperatures will rise above the historical average in Mexico. López-González et al. [4] and López-González et al. [5] by using SFA pointed out that the great global challenge is to increase food production through higher productivity.

**Competing interests:** The authors have declared that no competing interests exist.

Despite the mentioned advantages for SFA, this method is sensitive to a priori assumptions and requires a pre-specification of the functional form. In the meantime, Data Envelopment Analysis (DEA) is a non-parametric method that measures the efficiency of homogenous Decision-Making Units (DMUs) without needing any specification of the functional form of the production function. On the other hand, DEA is easy to implement that can utilize multiple inputs and multiple outputs simultaneously. Although the main purpose of DEA is to evaluate the efficiency of DMUs, it can also be used for other purposes such as solving problems of decision-making, management, and economics. Estimation of the congestion is one of these problems. Congestion is one of the important topics in data envelopment analysis. A DMU exhibits congestion if an increase (decrease) in one or more input(s) of the DMU leads to a decrease (increase) in one or more of its output(s). Congestion can be considered as a useless stage of the production process that reflects the problem of excessive inputs. In other words, congestion is a special type of inefficiency that is different from a well-known concept of inefficiency called "technical inefficiency" (see [6]). Therefore, identification and elimination of congestion are very important to increase efficiency or reduce costs.

Various studies have been conducted to identify and detect the congestion in the DEA framework. The concept of congestion was first investigated by Färe and Svensson [7] to introduce three forms of the concept of congestion for a production function with a single output. Then, Färe and Grosskopf [8] and Färe et al. [9] introduced a radial DEA method to compute the congestion effect by taking the ratio of the observed values to the expected values. It should be noted that their model shows only the existence or non-existence of congestion. Later, Cooper et al. [10] proposed a slack-based DEA approach to calculate the congestion by considering the difference between the observed values and the expected values. This approach determines the congested inputs and provides a measure for the value of congestion in each input. In this context, Cooper et al. [11] developed a necessary and sufficient condition for the presence of (input) congestion. They also proposed a unified additive model for evaluating congestion. Moreover, Cooper et al. [12] proposed a one-model to evaluate the congestion. Then, Jahanshahloo and Khodabakhshi [13] pointed out that reducing some inputs such as labors may be made to tension in society. Accordingly, they proposed a DEA congestion method to resolve this problem. Furthermore, Tone and Saho [14] suggested a method in a non-parametric framework to measure scale elasticity in production in the presence of congestion. However, their method was incapable of detecting congestion status in the presence of alternative optimal solutions. Sueyoshi and Sekitani [15] proposed a method to detect the congestion in the presence of multiple solutions. Wie and Yan [16] studied the problems of congestion using the DEA output-oriented models. Further studies in this field can be found in the work of Khodabakhshi et al. [17].

According to the aforementioned studies, the concept of congestion has been widely developed in recent years. For example, Adimi et al. [18] introduced the concept of congestion hyperplane without considering the efficiency value. Khoveyni et al. [19] proposed an integer-valued slack-based DEA approach for recognizing the right- and left-hand congestion status of the DMUs which are all characterized by the technology dealing with both negative and/or non-negative continuous and integer data. Shabanpour et al. [20] showed that an increase in congested inputs may lead to higher outputs/efficiency. They also used the concept of input congestion as a tool for ranking decision making units. In the context of resources saving and climate change, Chen [21] extended the Cooper-family model by using a range adjusted measure (RAM) approach to incorporate carbon emissions under the DEA framework. Xian-ton Ren et al. [22] tried to eliminate congestion by increasing input on research and development activities at Chinese universities. They also analyzed the relationship between congestion and

overinvestment. Navidi et al. [23] proposed the method that measures the congestion without solving a model. This method can be used for different Production Possibility Set like $T_{New}$ and FDH, and, different data like negative data and integer data. Cho and Yang [24] developed a new method for congestion analysis that keeps a close link between undesirable outputs, desirable outputs, and inputs. Shadab et al. [25] developed an algorithm by the connection between the anchor points and congestion definition. Khoshroo et al. [26] applied the bounded adjusted measure (BAM) for improving the efficiency of tomato production as well as decreasing the carbon footprint. They investigated the performance of tomato farms in Iran's provinces by using a DEA-BAM methodology to determine the efficient or inefficient tomato farms and suggest inefficiency sources. By using data envelopment analysis and productivity measures obtained via Malmquist index, Velázquez and Benita [27] investigated the patterns and dynamics of efficiency, productivity, and technological change of the automotive sector in Mexico.

Despite the availability of extensive studies on congestion, it should be noted that most of them have not paid attention to the internal structure of the decision making units. However, in practice, there are systems composed of independent production sub-units. Conventional DEA views such a system as a black-box and ignores the internal structures of DMUs. Accordingly, the existing methods often evaluate the congestion without considering the internal structures of DMUs.

A parallel system is one of the basic types of network structure in DEA in which all sub-units may be considered homogenous (with the same type of inputs and outputs) without any link. There are several studies to deal with the parallel network structure. For example, Kao [28] developed a parallel DEA model which takes the operation of individual components into account in calculating the efficiency of the system. To addresses the problem with the conventional DEA for not considering the internal structure, Bi et al. [29] proposed to divide the production activities within a DMU into two subsets or units. The first unit is termed as the core business unit, which includes the main production functions of DMU; the second unit is referred to as the non-core business unit. Xiong et al. [30] extended the DEA model to consider the one-sided heterogeneous problem in a multi-function parallel structure, handling subunit sets that have heterogeneity in outputs. An et al. [31] proposed an additive DEA model to measure a parallel interdependent processes system with two components that have an interdependent relationship. Xiong et al. [32] proposed a parallel DEA-based approach to reallocate multi-period resources among all DMUs by treating individual periods (e.g., years) as divisions operating at the parallel level. Wen et al. [33] constructed a cooperative game with coalition structures, named the DEA game with coalition structures. They viewed groups as homogeneous parallel DMUs and their subsidiaries as sub-units. Liu et al. [34] proposed the neutral cross-efficiency evaluation method for general parallel systems. They first developed the self-evaluation models for parallel system and its production-units. In this way, the model can enable each production-unit to participate in the evaluation effectively. Then, the neutral cross-efficiency model was proposed to overcome the defects of self-evaluation model. They proposed parallel DEA-based methods and the results show that if the non-existent outputs are replaced with zeros or missing values will lead to overestimate the efficiency of the DMU. Lu et al. [35] employed dynamic three-stage network data envelopment analysis (DEA), considering parallel production in the agricultural and industrial sectors, to assess the impact of greenhouse gas emissions on the climate change and natural disaster stages. Storto [36] carried out the efficiency analysis of the Italian urban water industry, employing an extended parallel network DEA model that allows a more comprehensive overview of the industry performance.

Ignoring the internal structure of the DMUs may lead to inadequate or even incorrect results. For this reason, in this paper, we are going to identify and evaluate the congestion assuming the existence of parallel process systems. To this end, the concept of congestion is defined based on the Production Possibility Set (PPS) corresponding to the mentioned parallel systems. Then, a one-model linear programming problem is proposed to identify and evaluate the overall congestion of DMUs. It is worth mentioning that the proposed model can also detect the congestion of sub-units.

The rest of the paper is organized as follows: Section 2 provides the required concepts and definitions, along with a brief description of the single-model method proposed by Cooper et al. [12]. In Section 3, a new one-model is proposed to detect and evaluate the congestion of the parallel processes system. In Section 4, the proposed model is illustrated using a numerical example and case study. In Section 5, the results of the proposed method are compared with the results of the existing black-box approach. Finally, the conclusion is summarized in Section 6.

## 2 Preliminaries

As mentioned, congestion is a special type of inefficiency that is different from technical inefficiency. In this section, the definition of input congestion and technical inefficiency is reviewed, and the difference between them is emphasized. Furthermore, the parallel processes system is introduced and investigated.

### 2.1 Classic input congestion

An important point about congestion is its difference from the concept of technical inefficiency. To clarify the issue, the definition of both concepts should be noted.

**Definition 2.1** (*Input Congestion*). *Input congestion occurs whenever the increase of one/more inputs decreases some outputs without improving other inputs or outputs. Conversely, congestion occurs when decreasing some of the inputs increases some outputs without worsening other inputs or outputs* [37].

**Definition 2.2** (*Technical Inefficiency*). *Technical inefficiency is present when it is possible to improve some inputs or outputs without worsening other inputs or outputs* [37].

According to Definition 2.2, in the situation of technical inefficiency, improvements may be made without the need to use more resources or further benefits in the form of outputs' reduction. However, based on Definition 2.1, in the presence of congestion, improvement in one or more outputs (without worsening other inputs or outputs) is achieved by reducing the congesting inputs. In other words, technical inefficiency represents an excess of some inputs or a shortfall in some output; but, when congestion is present, reductions in technical inefficiency are accompanied by output improvement [38]. This difference can easily be seen in Fig 1. Consider the production possibility set in Fig 1 including DMUs *A*, *B*, *C*, *D*, *E*, *F* and *G* with a single input and single output.

Each of the three DMUs *D*, *F*, and *G* are inefficient. Because there are some DMUs in the PPS (such as DMU *B*) that produce more output (at least in one component) by using less input (at least in one component). However, the only units that exhibit congestion are DMUs *D* and *G*. This is because the output of DMU *F* can not be increased by reducing its input. On the other hand, it should be noted that the output increase available at DMU *F* is distinguished from the maximally possible output increase available at DMU *D*. This can be accomplished by noting that congestion is a frontier concept that occurs when an increase in one or more input components is associated with the decrease that is maximally

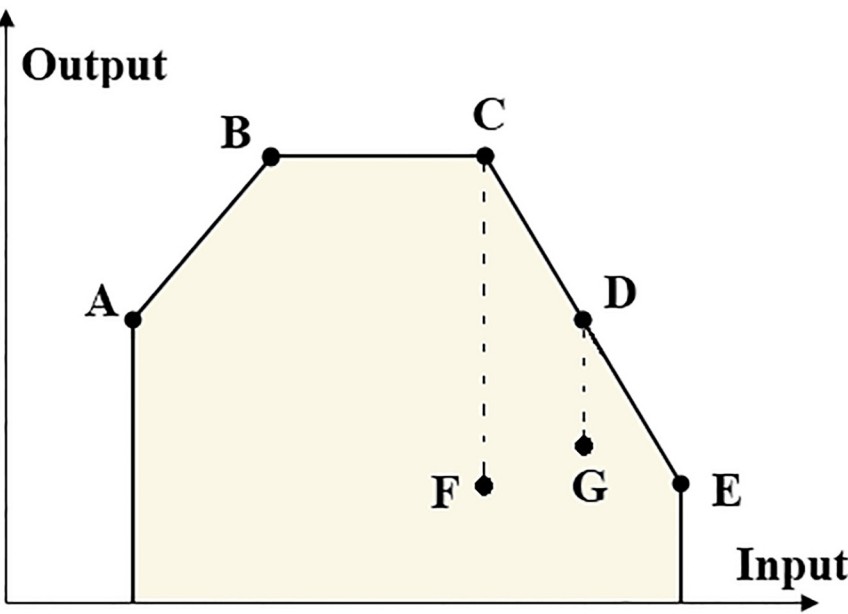

**Fig 1. Difference between congestion and inefficiency.**

possible in one or more output components without worsening other input or output components [38].

Another point is that congestion refers to a production possibility set in which the principle of input possibility is not present. In other words, the concept of input congestion is discussed on the production possibility set that is constructed based on the following principles:

**Principle 1 (Inclusion of observations).** According to this principle, all observed DMUs must be members of the PPS.

**Principle 2 (Convexity).** If $(X_1, Y_1)$ and $(X_2, Y_2)$ belong to the PPS then $\lambda(X_1, Y_1) + (1 - \lambda)(X_2, Y_2)$ belongs to the PPS, too; where, $0 \leq \lambda \leq 1$.

**Principle 3 (Output disposability).** If $(\bar{X}, \bar{Y})$ belongs to the PPS and $Y \leq \bar{Y}$, then $(\bar{X}, Y)$ belongs to the PPS, too.

**Principle 4 (Minimum interpolation).** PPS is the intersection set of all the sets satisfying Principle 1, 2 and 3.

Therefore, by considering the observed $DMU_j = (X_j, Y_j)$ $(j = 1, \ldots, n)$ that produces the output vector $Y_j$ by using the input vector $X_j$, the desired PPS satisfying Principle 1 to 4 is as follows:

$$T = \left\{ (X, Y) \mid \sum_{j=1}^{n} \lambda_j X_j = X, \ \sum_{j=1}^{n} \lambda_j Y_j \geq Y, \ \sum_{j=1}^{n} \lambda_j = 1, \ \lambda_j \geq 0 \ , j = 1, ..., n \right\} \tag{1}$$

There are several methods for identifying and evaluating the congestion based on the PPS (1). One of them is the proposed methods by Cooper et al. [12]. They developed a single-model method that combines the two models of the BCSW method [39]. The proposed single-model by Cooper et al. [12] to detect and evaluate the congestion of $DMU_o$ is as the following

model:

$$
\begin{aligned}
Max \quad & \varphi + \varepsilon\left(\sum_{r=1}^{s} s_r^+ - \varepsilon\sum_{i=1}^{m} s_i^{-c}\right) \\
s.t. \quad & \sum_{j=1}^{n}\lambda_j x_{ij} + s_i^{-c} = x_{io} \qquad i = 1, ..., m, \\
& \sum_{j=1}^{n}\lambda_j y_{rj} - s_r^+ = \varphi y_{ro} \qquad r = 1, ..., s, \\
& \sum_{j=1}^{n}\lambda_j = 1, \quad \lambda_j \geq 0 \qquad j = 1, ..., n, \\
& s_i^{-c}, s_r^+ \geq 0 \qquad i = 1, ..., m, \; r = 1, ..., s.
\end{aligned}
\tag{2}
$$

where, $X_j = (x_{1j}, ..., x_{ij}, ..., x_{mj})^t$, $Y_j = (y_{1j}, ..., y_{rj}, ..., y_{sj})^t$ and $\varepsilon$ is a small positive non-Archimedean value. It should be noted that the presence of $\varepsilon$ in Model (2) indicates the optimizing priority of the variables in the objective function. In other words, $\varphi$ is the first variable which should be maximized. Then the variables $s_r^+$ ($r = 1, ..., s$) are maximized and finally the variables $s_i^{-c}$ ($i = 1, ..., m$) are minimized.

**Theorem 2.1** *$DMU_o$ exhibits input congestion when at least one of the following conditions is satisfied for the optimal solution of Model (2), denoted by ($\varphi^*$, $\lambda^*$, $S^{-c^*}$, $S^{+^*}$):*

1. $\varphi^* > 1$ and $\sum_{i=1}^{m} s_i^{-c*} > 0$;

2. $\sum_{r=1}^{s} s_r^{+*} > 0$ and $\sum_{i=1}^{m} s_i^{-c*} > 0$.

*In this case, the optimal values $s_i^{-c*}$ ($i = 1, ..., m$) represent the value of inputs congestion* [12].

**Theorem 2.2** *$DMU_o$ is inefficient if at least one of the following conditions is satisfied:*

1. $\varphi^* > 1$;

2. $\sum_{r=1}^{s} s_r^{+*} > 0$;

3. $\sum_{i=1}^{m} s_i^{-c*} > 0$.

*where ($\varphi^*$, $\lambda^*$, $S^{-c^*}$, $S^{+^*}$) is an optimal solution of Model (2). Conversely, if $\varphi^* = 1$, $\sum_{r=1}^{s} s_r^{+*} = 0$ and $\sum_{i=1}^{m} s_i^{-c*} = 0$ then $DMU_o$ is on the (efficient or inefficient) frontier of production possibility set defined in relation (1)* [12].

## 2.2 Parallel network DEA and congestion

Network Data Envelopment Analysis (NDEA) uses the DEA technique to evaluate the performance of the decision making unit by considering its internal structure. In this way, the obtained results are more reliable than those obtained from the conventional DEA methods in which the DMUs are treated as a black-box. Two basic structures are considered in NDEA, i.e., series and parallel. These structures are the basis for general network structures. In the series structure, the subunits of a system are arranged in a sequence such that the outputs of one subunit are the inputs of the next. In this situation, a subunit can start its operation only

after its preceding subunits have finished their work. While, in the parallel system, all of the subunits appear in parallel and each subunit operates independently at the same time, without affecting each other. According to the function of the subunits, parallel systems can be classified into multi-component and multi-function systems. Multi-component systems are composed of several subunits with the same function. Each subunit uses the same inputs to produce the same outputs, and each DMU does not require the same number of divisions. In this case, each subunit can be compared not only within the same DMU but also among different DMUs. On the other hand, in the multi-function systems, each DMU has an equal number of subunits that perform a specific function. In this case, the subunits of a DMU are not homogenous and so, they cannot be compared with each other. However, the subunits of different DMUs of the same function are comparable.

In this paper, the concept of congestion is investigated for the multi-function parallel systems that have no share inputs/outputs. For a better understanding of these systems, consider Fig 2 that represents $DMU_j$ including $q$ subunits. Subunit $k$ ($k = 1, \ldots, q$) uses the inputs $x_{ij}^{(k)}$, $i = m^{(k-1)} + 1, \ldots, m^{(k-1)} + m^{(k)}$, to produce the outputs $y_{rj}^{(k)}$, $r = s^{(k-1)} + 1, \ldots, s^{(k-1)} + s^{(k)}$, where, $m^{(0)} = s^{(0)} = 0$ and $m^{(q)} = m$, $s^{(q)} = s$. It should be noted that using the superscript $(k)$ in $x_{ij}^{(k)}$ and $y_{rj}^{(k)}$ is not necessary but it is used to better identify the subunits.

Kao [40] proposed the following DEA model to measure the relative efficiency of $DMU_o$ with the aforementioned multi-function parallel system:

$$
\begin{aligned}
max \quad & \sum_{k=1}^{q} \sum_{r=s^{(k-1)}+1}^{s^{(k-1)}+s^{(k)}} u_r y_{ro}^{(k)} \\
s.t. \quad & \sum_{k=1}^{q} \sum_{i=m^{(k-1)}+1}^{m^{(k-1)}+m^{(k)}} v_i x_{io}^{(k)} = 1, \\
& \sum_{r=s^{(k-1)}+1}^{s^{(k-1)}+s^{(k)}} u_r y_{rj}^{(k)} - \sum_{i=m^{(k-1)}+1}^{m^{(k-1)}+m^{(k)}} v_i x_{ij}^{(k)} \le 0, \quad k = 1, \ldots, q, \; j = 1, \ldots, n \\
& u_r, v_i \ge \varepsilon, r = 1, \ldots, s, \; i = 1, \ldots, m.
\end{aligned}
\tag{3}
$$

where $\varepsilon$ is is a small non-Archimedean quantity that prohibits any input/output factor to be ignored. (see [41] and [42]). Note that to measure the efficiency of the DMUs, Model (3) should be enumerated for n times, once for each DMU. Kao [40] showed that the dual of Model (3) can be written as Model (4), by omitting the non-Archimedean amount $\varepsilon$:

$$
\begin{aligned}
min \quad & \theta \\
s.t. \quad & \sum_{j=1}^{n} \lambda_j^{(k)} x_{ij}^{(k)} \le \theta x_{io}^{(k)}, \quad i = m^{(k-1)} + 1, \ldots, m^{(k-1)} + m^{(k)}, \; k = 1, \ldots, q \\
& \sum_{j=1}^{n} \lambda_j^{(k)} y_{rj}^{(k)} \ge y_{ro}^{(k)}, \quad r = s^{(k-1)} + 1, \ldots, s^{(k-1)} + s^{(k)}, \; k = 1, \ldots, q \\
& \lambda_j^{(k)} \ge 0, j = 1, \ldots, n, \; k = 1, \ldots, q.
\end{aligned}
\tag{4}
$$

To the best of our knowledge, there is no study on the congestion evaluation of the multi-function parallel systems. Indeed, to identify and evaluate the congestion of these systems, the conventional congestion DEA methods should be used ignoring their internal structure. For this reason, in the next section, we are going to evaluate the congestion of the multi-function parallel systems

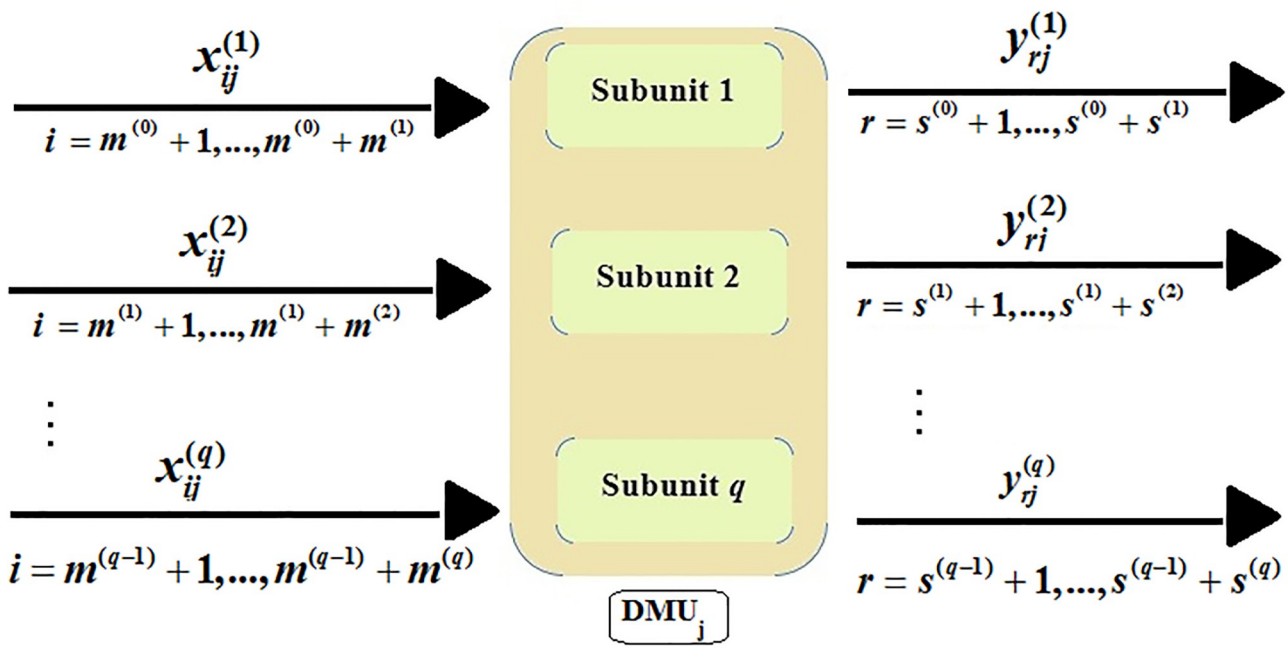

**Fig 2. Structure of the multi-function parallel system.**

by considering their internal structure. In other words, a method is proposed that detects and evaluates the congestion of the system as a whole as well as the congestion of its subunits.

## 3 Congestion of the multi-function parallel systems

Consider $n$ observed $DMU_j$ ($j = 1, \ldots, n$) with the $q$ parallel subunits, as shown in Fig 2. To discuss the congestion, first of all, the proper production possibility set should be defined corresponding to the production technology. Similar to the PPS proposed by Bi et al. [29], by considering the principles of observations' inclusion, convexity, input (According to the principle of input disposability, if $(\bar{X}, \bar{Y})$ belongs to the PPS and $X \geq \bar{X}$, then $(X, \bar{Y})$ belongs to the PPS, too.) and output disposability, and the principle of minimum interpolation, the PPS of the DMU as a whole can be defined as the relation (5):

$$T = \{(X^{(1)}, \ldots, X^{(q)}, Y^{(1)}, \ldots, Y^{(q)}) | \ (X^{(k)}, Y^{(k)}) \in T^{(k)}, k = 1, \ldots, q \} \tag{5}$$

where, $X^{(k)} = (x^{(k)}_{m^{k-1}+1}, \ldots, x^{(k)}_{m^{(k-1)}+m^k})$, $Y^{(k)} = (y^{(k)}_{s^{k-1}+1}, \ldots, y^{(k)}_{s^{(k-1)}+s^k})$ and $m^{(0)} = s^{(0)} = 0, m^{(q)} = m, s^{(q)} = s$; furthermore, $T^{(k)}$ ($k = 1, \ldots, q$) indicates the PPS of Subunit $k$ as the relation (6):

$$T^{(k)} = \{(X, Y) | \ \sum_{j=1}^{n} \lambda^{(k)}_j X^{(k)}_j \leq X, \ \sum_{j=1}^{n} \lambda^{(k)}_j Y^{(k)}_j \geq Y, \ \sum_{j=1}^{n} \lambda^{(k)}_j = 1, \ \lambda^{(k)}_j \geq 0, \ j = 1, \ldots, n \} \tag{6}$$

where $X = (x_{m^{k-1}+1}, \ldots, x_{m^{(k-1)}+m^k})$, $Y = (y_{s^{k-1}+1}, \ldots, y_{s^{(k-1)}+s^k})$ and $X^{(k)}_j$ along with the $Y^{(k)}_j$ ($j = 1, \ldots, n$) are the input and output vectors of Subunit $k$ in $DMU_j$. By eliminating the principle of input disposability, to detect and evaluate the congestion, the production possibility sets $T$ and $T^{(k)}$ are changed as the relations (7) and (8), respectively:

$$T^c = \{(X^{(1)}, \ldots, X^{(q)}, Y^{(1)}, \ldots, Y^{(q)}) | \ (X^{(k)}, Y^{(k)}) \in T^{c,(k)}, k = 1, \ldots, q \} \tag{7}$$

$$T^{c,(k)} = \left\{ (X, Y)\,\middle|\; \sum_{j=1}^{n}\lambda_j^{(k)}X_j^{(k)} = X,\;\; \sum_{j=1}^{n}\lambda_j^{(k)}Y_j^{(k)} \geq Y,\;\; \sum_{j=1}^{n}\lambda_j^{(k)} = 1,\;\; \lambda_j^{(k)} \geq 0,\;\; j = 1,...,n \right\}$$ (8)

Now, by having the production possibility sets $T^c$ and $T^{c,(k)}$, the concept of the congestion can be defined for a DMU and its subunits with the mentioned structure.

**Definition 3.1** $DMU_o = (X_o^{(1)}, \ldots, X_o^{(q)}, Y_o^{(1)}, \ldots, Y_o^{(q)})$ *with the multi-function parallel network structure, as shown in* Fig 2, *exhibits congestion as a whole if increasing (decreasing) of one/more inputs $x_{io}^{(k)}$ s ($k = 1, \ldots, q, i = m^{(k-1)}, \ldots, m^{(k-1)} + m^{(k)}$) decreases (increases) one/more outputs $y_{ro}^{(k)}$ s ($k = 1, \ldots, q, r = s^{(k-1)}, \ldots, s^{(k-1)} + s^{(k)}$) in $T^c$, without improving (worsening) other input or output components.*

**Definition 3.2** *Subunit k of $DMU_o$, i.e., $(X_o^{(k)}, Y_o^{(k)})$, with the parallel network structure, as shown in* Fig 2, *exhibits congestion if increasing (decreasing) of one/more input components of $X_o^{(k)}$ decreases (increases) one/more output components of $Y_o^{(k)}$ in $T^{c,(k)}$, without improving (worsening) other input or output components.*

Therefore, considering the structure of $T^c$ and Definition 3.1, Model (9) is proposed to detect and evaluate the congestion of $DMU_o$ as a whole:

$$
\begin{aligned}
Max \quad & \frac{1}{q}\sum_{k=1}^{q}\varphi_o^{(k)} + \varepsilon\left(\sum_{k=1}^{q}\sum_{r=s^{(k-1)}+1}^{s^{(k-1)}+s^{(k)}}d_r^{+,(k)} - \varepsilon\sum_{k=1}^{q}\sum_{i=m^{(k-1)}+1}^{m^{(k-1)}+m^{(k)}}d_i^{-c,(k)}\right) \\
s.t. \quad & \sum_{j=1}^{n}\lambda_j^{(k)}x_{ij}^{(k)} + d_i^{-c,(k)} = x_{io}^{(k)},\;\; i = m^{(k-1)}+1,...,m^{(k-1)}+m^{(k)},\;\; k = 1,...,q, \\
& \sum_{j=1}^{n}\lambda_j^{(k)}y_{rj}^{(k)} - d_r^{+,(k)} = \varphi_o^{(k)}y_{ro}^{(k)},\;\; r = s^{(k-1)}+1,...,s^{(k-1)}+s^{(k)},\;\; k = 1,...,q, \\
& \sum_{j=1}^{n}\lambda_j^{(k)} = 1,\quad k = 1,...,q, \\
& d_i^{-c,(k)},\;\; k = 1,...,q,\;\; i = m^{(k-1)}+1,...,m^{(k-1)}+m^{(k)}, \\
& d_r^{+,(k)} \geq 0,\;\; k = 1,...,q,\;\; r = s^{(k-1)}+1,...,s^{(k-1)}+s^{(k)} \\
& \lambda_j^{(k)} \geq 0, j = 1,...,n,\;\; k = 1,...,q.
\end{aligned}
$$ (9)

Of course, similar to Model (2), the presence of $\varepsilon$ indicates the optimizing priority of the variables in the objective function. The next point is the separability of the Model (9). In other words, since the constraints' variables of each subunit are independent of the constraints' variables of the other subunits, to solve this model, it is sufficient to solve the problem for each subunit independently. In this way, Model (10) can be used to detect and evaluate the congestion of the $p$-th subunit of $DMU_o$, which is also consistent with the definition of $T^{c,(p)}$ and Defini-

tion 3.2:

$$
Max \quad \varphi_o^{(p)} + \varepsilon \left( \sum_{r=s^{(k-1)}+1}^{s^{(p-1)}+s^{(p)}} d_r^{+,(p)} - \varepsilon \sum_{i=m^{(p-1)}+1}^{m^{(p-1)}+m^{(p)}} d_i^{-c,(p)} \right)
$$

$$
s.t. \quad \sum_{j=1}^{n} \lambda_j^{(p)} x_{ij}^{(p)} + d_i^{-c,(p)} = x_{io}^{(p)}, \quad i = m^{(p-1)} + 1, \ldots, m^{(p-1)} + m^{(p)},
$$

$$
\sum_{j=1}^{n} \lambda_j^{(p)} y_{rj}^{(p)} - d_r^{+,(p)} = \varphi_o^{(p)} y_{ro}^{(p)}, \quad r = s^{(p-1)} + 1, \ldots, s^{(p-1)} + s^{(p)}, \tag{10}
$$

$$
\sum_{j=1}^{n} \lambda_j^{(p)} = 1,
$$

$$
d_i^{-c,(p)}, \quad i = m^{(p-1)} + 1, \ldots, m^{(p-1)} + m^{(p)},
$$
$$
d_r^{+,(p)} \geq 0, \quad r = s^{(p-1)} + 1, \ldots, s^{(p-1)} + s^{(p)},
$$
$$
\lambda_j^{(p)} \geq 0, j = 1, \ldots, n.
$$

**Remark 3.1** *Both Models* (9) *and* (10) *are linear programming problems that can be easily solved with software like General Algebraic Modeling System (GAMS) and MATLAB. In better words, problems such as local optimality will not occur in solving these models and the existing software can reach the optimal solution in just a few repetitions of its used algorithms (e.g., simplex).*

**Lemma 3.1** *The value of the* $\varphi_o^{(k)}$ *is greater than or equal to 1, for all* $k = 1, \ldots, q,$ *in any optimal solution of Model* (9).

**Proof.** Assume (by contradiction) that $(\Phi_o^*, \Lambda^*, D^{-c*}, D^{+*}) = (\Phi_o^*, \Lambda^{(1)*}, \ldots, \Lambda^{(q)*}, D^{-c,(1)*}, \ldots D^{-c,(q)*}, D^{+,(1)*}, \ldots D^{+,(q)*}$ is an optimal solution of Model (9) in which:

$$
\Phi_o^* = (\varphi_o^{(1)*}, \ldots, \varphi_o^{(q)*}),
$$
$$
\Lambda^{(k)*} = (\lambda_1^{(k)*}, \ldots, \lambda_n^{(k)*}), \quad k = 1, \ldots, q,
$$
$$
D^{+,(k)*} = (d_{s^{(k-1)}+1}^{+,(k)*}, \ldots, d_{s^{(k-1)}+s^{(k)}}^{+,(k)*}), \quad k = 1, \ldots, q, \tag{11}
$$
$$
D^{-c,(k)*} = (d_{s^{(k-1)}+1}^{-c,(k)*}, \ldots, d_{s^{(k-1)}+s^{(k)}}^{-c,(k)*}), \quad k = 1, \ldots, q.
$$

and there exists $p \in \{1, \ldots, q\}$ such that $\varphi_o^{(p)*} < 1$. Now, consider $(\bar{\Phi}_o, \bar{\Lambda}, \bar{D}^{-c}, \bar{D}^+)$ as defined in the relation (12):

$$
\bar{\varphi}_o^{(k)} = \begin{cases} \varphi_o^{(k)*} & k = 1, \ldots, q, k \neq p, \\ 1 & k = p \end{cases},
$$

$$
\bar{\lambda}_j^{(k)} = \begin{cases} \lambda_j^{(k)*} & k = 1, \ldots, q, \ k \neq p, \ j = 1, \ldots, n \\ 1 & k = p, \ j = o \\ 0 & k = p, \ j = 1, \ldots, n, j \neq o \end{cases},
$$

$$
\bar{d}_r^{+,(k)} = \begin{cases} d_r^{+,(k)*} & r = s^{(k-1)} + 1, \ldots, s^{(k-1)} + s^{(k)}, \ k = 1, \ldots, q, k \neq p \\ 0 & r = s^{(k-1)} + 1, \ldots, s^{(k-1)} + s^{(k)}, \ k = p \end{cases}, \tag{12}
$$

$$
\bar{d}_i^{-c,(k)} = \begin{cases} d_i^{-c,(k)*} & i = m^{(k-1)} + 1, \ldots, m^{(k-1)} + m^{(k)}, \ k = 1, \ldots, q, k \neq p \\ 0 & i = m^{(k-1)} + 1, \ldots, m^{(k-1)} + m^{(k)}, \ k = p \end{cases},
$$

It is clear that $(\bar{\Phi}_o, \bar{\Lambda}, \bar{D}^{-c}, \bar{D}^+)$ is a feasible solution of Model (9) in which the value of the objective function is greater than the optimum value of the objective function (Since $\varepsilon$ is an

infinitesimal amount, the sentences containing the $\varepsilon$ can be omitted in calculating the value of the objective function.), and that is a contradiction.

**Theorem 3.1** $DMU_o = (X_o^{(1)}, \ldots, X_o^{(q)}, Y_o^{(1)}, \ldots, Y_o^{(q)})$ *with the parallel network structure, as shown in* Fig 2, *exhibits input congestion in* $T^c$ *if and only if in the optimal solution of Model* (9), *at least one of the following two conditions is satisfied*:

1. $\frac{1}{q}\sum_{k=1}^{q} \varphi_o^{(k)*} > 1$ *and* $\sum_{k=1}^{q} \sum_{i=m^{(k-1)}+1}^{m^{(k-1)}+m^{(k)}} d_i^{-c,(k)*} > 0$;

2. $\sum_{k=1}^{q} \sum_{r=s^{(k-1)}+1}^{s^{(k-1)}+s^{(k)}} d_r^{+,(k)*} > 0$ *and* $\sum_{k=1}^{q} \sum_{i=m^{(k-1)}+1}^{m^{(k-1)}+m^{(k)}} d_i^{-c,(k)*} > 0$.

*where, '\** *denotes the optimal solution. In this case,* $d_i^{-c,(k)*}$ *represents the congesting amount of the i-th input of* $DMU_o$ *in its k-th subunit.*

**Proof.** Suppose that $DMU_o$ exhibits a congestion according to Definition 3.1. Therefore, there is a unit such as $(\bar{X}, \bar{Y})$ in the production possibility set $T^c$ such that $\bar{X} \lneqq X_o, \bar{Y} \gneqq Y_o$. Thus, according to the membership condition of $T^c$ for $(\bar{X}, \bar{Y}) = (\bar{X}^{(1)}, \ldots, \bar{X}^{(q)}, \bar{Y}^{(1)}, \ldots, \bar{Y}^{(q)})$, there are $\bar{\lambda}_j^{(k)}$s $(j = 1, \ldots, n, k = 1, \ldots, q)$ that satisfy the constraints of the relation (13):

$$\sum_{j=1}^{n}\bar{\lambda}_j^{(k)} X_j^{(k)} = \bar{X}^{(k)} \leq X_o^{(k)}, \quad k = 1, \ldots, q,$$

$$\sum_{j=1}^{n}\bar{\lambda}_j^{(k)} Y_j^{(k)} = \bar{Y}^{(k)} \geq Y_o^{(k)}, \quad k = 1, \ldots, q,$$

$$\sum_{j=1}^{n}\bar{\lambda}_j^{(k)} = 1, \quad k = 1, \ldots, q, \tag{13}$$

$$\bar{\lambda}_j^{(k)} \geq 0, j = 1, \ldots, n, \ k = 1, \ldots, q.$$

Here, it should be noted that there exists at least one $g \in \{1, \ldots, q\}$ such that $\sum_{j=1}^{n} \bar{\lambda}_j^{(g)} X_j^{(g)} = \bar{X}^{(g)} \lneqq X_o^{(g)}$ and there exists at least one $p \in \{1, \ldots, q\}$ such that $\sum_{j=1}^{n} \bar{\lambda}_j^{(p)} Y_j^{(p)} = \bar{Y}^{(p)} \gneqq Y_o^{(p)}$. Now, by defining the slack variables $\bar{D}^{+,(k)} = (\bar{d}_{s^{(k-1)}+1}^{+,(k)}, \ldots, \bar{d}_{s^{(k-1)}+s^{(k)}}^{+,(k)})$ and $\bar{D}^{-c,(k)} = (\bar{d}_{s^{(k-1)}+1}^{-c,(k)}, \ldots, \bar{d}_{s^{(k-1)}+s^{(k)}}^{-c,(k)})$ corresponding to the first and second constraints of the relation (13), respectively, this relation can be rewritten as follows:

$$\sum_{j=1}^{n}\bar{\lambda}_j^{(k)} X_j^{(k)} + \bar{D}^{-c,(k)} = X_o^{(k)}, \quad k = 1, \ldots, q,$$

$$\sum_{j=1}^{n}\bar{\lambda}_j^{(k)} Y_j^{(k)} - \bar{D}^{+,(k)} = Y_o^{(k)}, \quad k = 1, \ldots, q,$$

$$\sum_{j=1}^{n}\bar{\lambda}_j^{(k)} = 1, \quad k = 1, \ldots, q, \tag{14}$$

$$\bar{D}^{-c,(k)} \geq 0, \ \bar{D}^{+,(k)} \geq 0,$$

$$\bar{\lambda}_j^{(k)} \geq 0, j = 1, \ldots, n, \ k = 1, \ldots, q.$$

where, there exists at least one $g \in \{1, \ldots, q\}$ such that $\bar{D}^{-c,(g)} \gneqq 0$ and there exists at least one $p \in \{1, \ldots, q\}$ such that $\bar{D}^{+,(k)} \gneqq 0$. According to the relation (14), $(\bar{\Phi}_o, \bar{\Lambda}^{(1)}, \ldots, \bar{\Lambda}^{(q)}, \bar{D}^{-c,(1)}, \ldots, \bar{D}^{-c,(q)}, \bar{D}^{+,(1)}, \ldots, \bar{D}^{+,(q)})$ is a feasible solution of Model (9),

where the relation (15) holds:

$$\bar{\Phi}_o = (\bar{\varphi}_o^{(1)}, \ldots, \bar{\varphi}_o^{(q)}) = (1, 1, \ldots, 1),$$
$$\bar{\Lambda}^{(k)} = (\bar{\lambda}_1^{(k)}, \ldots, \bar{\lambda}_n^{(k)}), \qquad k = 1, \ldots, q. \tag{15}$$

Since $\sum_{k=1}^{q} \sum_{r=s^{(k-1)}+1}^{s^{(k-1)}+s^{(k)}} \bar{d}_r^{+,(k)*} > 0$ and $\sum_{k=1}^{q} \sum_{i=m^{(k-1)}+1}^{m^{(k-1)}+m^{(k)}} \bar{d}_i^{-c,(k)*} > 0$, at least the second condition of Theorem 3.1 is satisfied, and thus the proof is complete.

Conversly, suppose that Model (9) has an optimal solution that satisfies the relation (16)

$$\sum_{k=1}^{q} \sum_{i=m^{(k-1)}+1}^{m^{(k-1)}+m^{(k)}} \bar{d}_i^{-c,(k)*} > 0 \tag{16}$$

with at least one of the conditions $\frac{1}{q} \sum_{k=1}^{q} \varphi_o^{(k)*} > 1$ or $\sum_{k=1}^{q} \sum_{r=s^{(k-1)}+1}^{s^{(k-1)}+s^{(k)}} \bar{d}_r^{+,(k)*} > 0$. In this case, according to the constraints of Model (9) in the optimal solution, relation (17) holds:

$$\sum_{j=1}^{n} \bar{\lambda}_j^{(k)*} X_j^{(k)} = X_o^{(k)} - D^{-c,(k)*}, \quad k = 1, \ldots, q,$$

$$\sum_{j=1}^{n} \bar{\lambda}_j^{(k)*} Y_j^{(k)} = \varphi_o^{(k)*} Y_o^{(k)} + D^{+,(k)*}, \quad k = 1, \ldots, q,$$

$$\sum_{j=1}^{n} \bar{\lambda}_j^{(k)*} = 1, \quad k = 1, \ldots, q, \tag{17}$$

$$D^{-c,(k)*} \geq 0, \ D^{+,(k)*} \geq 0, \quad k = 1, \ldots, q,$$

$$\bar{\lambda}_j^{(k)*} \geq 0, j = 1, \ldots, n, \ k = 1, \ldots, q.$$

where, $D^{-c,(g)*} \gneqq 0$ for at least one $g \in \{1, \ldots, q\}$ and also $D^{+,(p)*} \gneqq 0$ for at least one $p \in \{1, \ldots, q\}$. Therefore, according to the relation (17) and Lemma 3.1, $(X_o^{(1)} - D^{-c,(1)*}, \ldots, X_o^{(q)} - D^{-c,(q)*}, \varphi_o^{(1)*} Y_o^{(1)} + D^{+,(1)*}, \ldots, \varphi_o^{(q)*} Y_o^{(q)} + D^{+,(q)*})$ is a member of $T^c$ that using the less input than $X_o$ produces the greater output than $Y_o$. This means that $DMU_o$ exhibits congestion as a whole.

**Theorem 3.2** Subunit $(X_o^{(p)}, Y_o^{(p)})$ of $DMU_o = (X_o^{(1)}, \ldots, X_o^{(q)}, Y_o^{(1)}, \ldots, Y_o^{(q)})$ with the parallel network structure, as shown in Fig 2, exhibits input congestion in $T^{c,p}$ if and only if in the optimal solution of Model (10), denoted by $(\varphi_o^{(p)*}, \Lambda^{(p)*}, D^{-c,(p)*}, D^{+,(p)*})$, at least one of the following two conditions is satisfied:

1. $\varphi_o^{(p)*} > 1$ and $\sum_{i=m^{(p-1)}+1}^{m^{(p-1)}+m^{(p)}} d_i^{-c,(p)*} > 0$;

2. $\sum_{r=s^{(p-1)}+1}^{s^{(p-1)}+s^{(p)}} d_r^{+,(p)*} > 0$ and $\sum_{i=m^{(p-1)}+1}^{m^{(p-1)}+m^{(p)}} d_i^{-c,(p)*} > 0$.

where,
$\Lambda^{(p)*} = (\lambda_1^{(p)*}, \ldots, \lambda_n^{(p)*})^t$, $D^{-c,(p)*} = (d_1^{-c,(p)*}, \ldots, d_m^{-c,(p)*})^t$, $D^{+,(p)*} = (d_1^{+,(p)*} \ldots, d_s^{+,(p)*})$. In this case, $d_i^{-c,(p)*}$ represents the congesting amount of i-th input of Subunit p.

**Proof.** The proof is similar to the proof of Theorem 3.1.

**Lemma 3.2** *Suppose that in the optimal solution of Model* (10) $\sum_{i=m^{(p-1)}+1}^{m^{(p-1)}+m^{(p)}} d_i^{-c,(p)*} > 0$. *In this case,* $\varphi_o^{(p)*} > 1$ *or* $\sum_{r=s^{(p-1)}+1}^{s^{(p-1)}+s^{(p)}} d_r^{+,(p)*} > 0$.

**Proof.** *Assume (by contradiction) that* $\varphi_o^{(p)*} = 1$ *and* $\sum_{r=s^{(p-1)}+1}^{s^{(p-1)}+s^{(p)}} d_r^{+,(p)*} = 0$. *Then, since* $d_r^{+,(p)*} \geq 0$, *it is concluded that* $d_r^{+,(p)*} = 0$ *for all* $r = s^{(p-1)} + 1, \ldots, s^{(p-1)} + s^{(p)}$. *On the other hand, a feasible solution of Model* (10) *can be obtained by defining* $(\bar{\varphi}_o^{(p)}, \bar{\Lambda}^{(p)}, \bar{D}^{-c,(p)}, \bar{D}^{+,(p)})$ *as the relation* (18):

$$
\begin{aligned}
&\bar{\varphi}_o^{(p)} = 1, \\
&\bar{\Lambda}^{(p)} = (\bar{\lambda}_1^{(p)}, \ldots, \bar{\lambda}_n^{(p)}), \ \bar{\lambda}_o^{(p)} = 1, \bar{\lambda}_j^{(p)} = 0 \quad j = 1, \ldots, n, \ j \neq o, \\
&\bar{D}^{+,(p)} = (\bar{d}_{s^{(p-1)}+1}^{+,(p)}, \ldots, \bar{d}_{s^{(p-1)}+s^{(p)}}^{+,(p)}) = (0, \ldots, 0), \\
&\bar{D}^{-c,(p)} = (\bar{d}_{s^{(p-1)}+1}^{-c,(p)}, \ldots, \bar{d}_{s^{(p-1)}+s^{(p)}}^{-c,(p)}) = (0, \ldots, 0).
\end{aligned}
\tag{18}
$$

*In this way,* $(\tilde{\varphi}_o^{(p)}, \tilde{\Lambda}^{(p)}, \tilde{D}^{-c,(p)}, \tilde{D}^{+,(p)}) = \frac{1}{2}(\varphi_o^{(p)*}, \Lambda^{(p)*}, D^{-c,(p)*}, D^{+,(p)*}) + \frac{1}{2}(\bar{\varphi}_o^{(p)}, \bar{\Lambda}^{(p)}, \bar{D}^{-c,(p)}, \bar{D}^{+,(p)})$ *is a feasible solution of Model* (10) *that has a better objective value than the optimal solution. It is a contradiction and shows that* $\varphi_o^{(p)*} > 1$ *or* $\sum_{r=s^{(p-1)}+1}^{s^{(p-1)}+s^{(p)}} d_r^{+,(p)*} > 0$.

**Corollary 3.1** *Similar to Lemma 3.2, it can be proved that if in the optimal solution of Model* (9), $\sum_{k=1}^{q} \sum_{i=m^{(k-1)}+1}^{m^{(k-1)}+m^{(k)}} d_i^{-c,(k)*} > 0$, *then,* $\frac{1}{q} \sum_{k=1}^{q} \varphi_o^{(k)*} > 1$ *or* $\sum_{k=1}^{q} \sum_{r=s^{(k-1)}+1}^{s^{(k-1)}+s^{(k)}} d_r^{+,(k)*} > 0$.

**Theorem 3.3** $DMU_o = (X_o^{(1)}, \ldots, X_o^{(q)}, Y_o^{(1)}, \ldots, Y_o^{(q)})$ *exhibits congestion as a whole if and only if there exists at least one subunit such as* $(X_o^{(p)}, Y_o^{(p)})$ *that exhibits congestion.*

**Proof.** *Suppose that* $DMU_o$ *exhibits congestion as a whole. Then,* $\sum_{k=1}^{q} \sum_{i=m^{(k-1)}+1}^{m^{(k-1)}+m^{(k)}} d_i^{-c,(k)*} > 0$ *in the optimal solution of Model* (9) *i.e.,* $(\Phi_o^*, \Lambda^*, D^{-c*}, D^{+*})$. *Therefore, there exists at least one* $p \in \{1, \ldots, q\}$ *such that* $\sum_{i=m^{(p-1)}+1}^{m^{(p-1)}+m^{(p)}} d_i^{-c,(p)*} > 0$. *On the other hand,* $(\varphi_o^{(p)*}, \Lambda^{(p)*}, D^{-c,(p)*}, D^{+,(p)*})$ *is a feasible solution of Model* (10) *in which* $\varphi_o^{(p)*} \geq 1$, *according to Lemma 3.1. In this situation, similar to Lemma 3.2, it can be shown that* $\varphi_o^{(p)*} > 1$ *or* $\sum_{r=s^{(p-1)}+1}^{s^{(p-1)}+s^{(p)}} d_r^{+,(p)*} > 0$. *This shows that at least subunit* $(X_o^{(p)}, Y_o^{(p)})$ *exhibits congestion.*

*Conversely, suppose that* $(X_o^{(p)}, Y_o^{(p)})$ *exhibits congestion. Then, Model* (10) *has an optimal solution such as* $(\varphi_o^{(p)*}, \Lambda^{(p)*}, D^{-c,(p)*}, D^{+,(p)*})$ *in which* $\sum_{i=m^{(p-1)}+1}^{m^{(p-1)}+m^{(p)}} d_i^{-c,(p)*} > 0$ *and at least one of the conditions* $\varphi_o^{(p)*} > 1$ *or* $\sum_{r=s^{(p-1)}+1}^{s^{(p-1)}+s^{(p)}} d_r^{+,(p)*} > 0$ *is satisfied. Accordingly, consider* $(\bar{\Phi}_o, \bar{\Lambda}, \bar{D}^{-c}, \bar{D}^+)$ *as defined in the relation* (19):

$$
\bar{\varphi}_o^{(k)} = \begin{cases} 1 & k = 1, \ldots, q, k \neq p \\ \varphi_o^{(p)*} & k = p \end{cases},
$$

$$\bar{\lambda}_j^{(k)} = \begin{cases} 1 & k = 1, ..., q, \ k \neq p, \ j = o \\ 0 & k = 1, ..., q, \ k \neq p, \ j = 1, ..., n, j \neq o, \\ \lambda_j^{(p)*} & j = 1, ..., n, \ k = p \end{cases}$$

$$\bar{d}_r^{+,(k)} = \begin{cases} 0 & r = s^{(k-1)} + 1, ..., s^{(k-1)} + s^{(k)}, \ k = 1, ..., q, k \neq p \\ d_r^{+,(p)*} & r = s^{(k-1)} + 1, ..., s^{(k-1)} + s^{(k)}, \ k = p \end{cases}, \tag{19}$$

$$\bar{d}_i^{-c,(k)} = \begin{cases} 0 & i = m^{(k-1)} + 1, ..., m^{(k-1)} + m^{(k)}, \ k = 1, ..., q, k \neq p \\ d_i^{-c,(p)*} & i = m^{(k-1)} + 1, ..., m^{(k-1)} + m^{(k)}, \ k = p \end{cases},$$

*Relation* (19) *shows that* $(\bar{\Phi}_o, \bar{\Lambda}, \bar{D}^{-c}, \bar{D}^+)$ *is a feasible solution of Model* (9) *such that* $\sum_{k=1}^{q} \sum_{i=m^{(k-1)}+1}^{m^{(k-1)}+m^{(k)}} \bar{d}_i^{-c,(k)}$ *is positive and at least one of the conditions* $\frac{1}{q} \sum_{k=1}^{q} \bar{\varphi}_o^{(k)} > 1$ *or* $\sum_{k=1}^{q} \sum_{r=s^{(k-1)}+1}^{s^{(k-1)}+s^{(k)}} \bar{d}_r^{+,(k)} > 0$ *is satisfied. Therefore, it is clear that Model* (9) *has an optimal solution such as* $(\Phi_o^{**}, \Lambda^{**}, D^{-c**}, D^{+**})$ *in which* $\sum_{k=1}^{q} \sum_{i=m^{(k-1)}+1}^{m^{(k-1)}+m^{(k)}} \bar{d}_i^{-c,(k)**} > 0$ *and at least one of the conditions* $\frac{1}{q} \sum_{k=1}^{q} \bar{\varphi}_o^{(k)**} > 1$ *or* $\sum_{k=1}^{q} \sum_{r=s^{(k-1)}+1}^{s^{(k-1)}+s^{(k)}} \bar{d}_r^{+,(k)**} > 0$ *is satisfied. This means that* $DMU_o$ *exhibits congestion as whole and the proof is complete.*

**Theorem 3.4** *Suppose that* $DMU_o = (X_o^{(1)}, ..., X_o^{(q)}, Y_o^{(1)}, ..., Y_o^{(q)})$ *exhibits congestion according to the single-model* (2) *when considered as black-box* $(X_o, Y_o)$. *Then,* $DMU_o$ *also exhibits congestion according to the proposed model* (9).

**Proof**. *To prove this theorem, before any things, it is sufficient to note that using the superscript* $(k)$ *in* $x_{ij}^{(k)}$ *and* $y_{rj}^{(k)}$ *was not necessary. This is also true for the superscript of the* $d_i^{-c,(k)}$ *s and* $d_r^{+,(k)}$. *Therefore, the superscript* $(k)$ *can be eliminated from the* $x_{ij}^{(k)}$ *s,* $y_{rj}^{(k)}$ *s,* $d_i^{-c,(k)}$ *s and* $d_r^{+,(k)}$ *in the model* (9). *Now, by setting* $\varphi_o^{(k)} = \varphi_o \ (k = 1, ..., q)$ *and* $\lambda_j^{(k)} = \lambda_j \ (k = 1, ..., q, j = 1, ..., n)$, *Model* (9) *is clearly transformed to Model* (2). *This shows that any feasible solution of Model* (2) *is also a feasible solution of Model* (9). *In this way, it is clear that if* $DMU_o$ *exhibits congestion according to the single-model* (2) *then it also exhibits congestion according to Model* (9).

**Remark 3.2** *The converse of Theorem 3.4 is not necessarily true. Indeed,* $DMU_o$ *may exhibit congestion according to Model* (9), *while it does not exhibit congestion according to the single-model* (2) (This will be shown in the next section). *This shows that the proposed model* (9) *can correctly generalize the concept of congestion from the black-box view to the multi-function parallel network view.*

## 4 Numerical example and case study

In this section, the proposed definitions and models are illustrated and investigated using a numerical example and a real case study.

## 4.1 Numerical example

Here, the proposed model (9) is applied to analyze the congestion of 8 hypothetical (unreal) education institutions. These institutions are composed of two independent subunits, namely Teaching and Research Subunits. We define the input and output components of each subunit as shown in Fig 3.

The number of the students and teachers are considered as two inputs of the teaching subunit, and its only output is the number of the students admitted to the final exam of the institution. Moreover, the amount of the research grants and the number of the valid publications of each institution are considered as single input and output of its research subunit, respectively. Table 1 shows the data set corresponding to these 8 institutions.

Consider Institution 1 as an example. Based on Model (3), to identify the congestion of Institution 1 as black-box, Model (20) can be used as follows:

$$
\begin{aligned}
Max \quad & \varphi + \varepsilon\left(\sum_{r=1}^{2} s_r^+ - \varepsilon\sum_{i=1}^{3} s_i^{-c}\right) \\
s.t. \quad & \sum_{j=1}^{8} \lambda_j x_{ij} + s_i^{-c} = x_{i1} \qquad i = 1, 2, 3, \\
& \sum_{j=1}^{8} \lambda_j y_{rj} - s_r^+ = \varphi y_{r1} \qquad r = 1, 2, \\
& \sum_{j=1}^{8} \lambda_j = 1, \quad \lambda_j \geq 0 \qquad j = 1, \ldots, 8, \\
& s_i^{-c}, s_r^+ \geq 0 \qquad i = 1, 2, 3, \ r = 1, 2.
\end{aligned}
\tag{20}
$$

where, $x_{ij} = x_{ij}^{(k)}$ and $y_{rj} = y_{rj}^{(k)}$. Note that Model (20) can be written in the equivalent form as Model (21):

$$
\begin{aligned}
Max \quad & \varphi + \varepsilon(d_1^{+,(1)} + d_2^{+,(2)} - \varepsilon(d_1^{-c,(1)} + d_2^{-c,(1)} + d_3^{-c,(2)})) \\
s.t. \quad & \sum_{j=1}^{8} \lambda_j x_{ij}^{(1)} + d_i^{-c,(1)} = x_{i1}^{(1)}, \quad i = 1, 2, \\
& \sum_{j=1}^{8} \lambda_j x_{3j}^{(2)} + d_3^{-c,(2)} = x_{31}^{(2)}, \\
& \sum_{j=1}^{8} \lambda_j y_{1j}^{(1)} - d_1^{+,(1)} = \varphi y_{11}^{(1)}, \\
& \sum_{j=1}^{8} \lambda_j y_{2j}^{(2)} - d_2^{+,(2)} = \varphi y_{21}^{(2)}, \\
& \sum_{j=1}^{n} \lambda_j = 1, \\
& d_i^{-c,(p)}, d_r^{+,(p)} \geq 0, \quad i = 1, 2, 3, \ r = 1, 2, \ p = 1, 2, \\
& \lambda_j \overset{\geq}{} 0, j = 1, \ldots, 8.
\end{aligned}
\tag{21}
$$

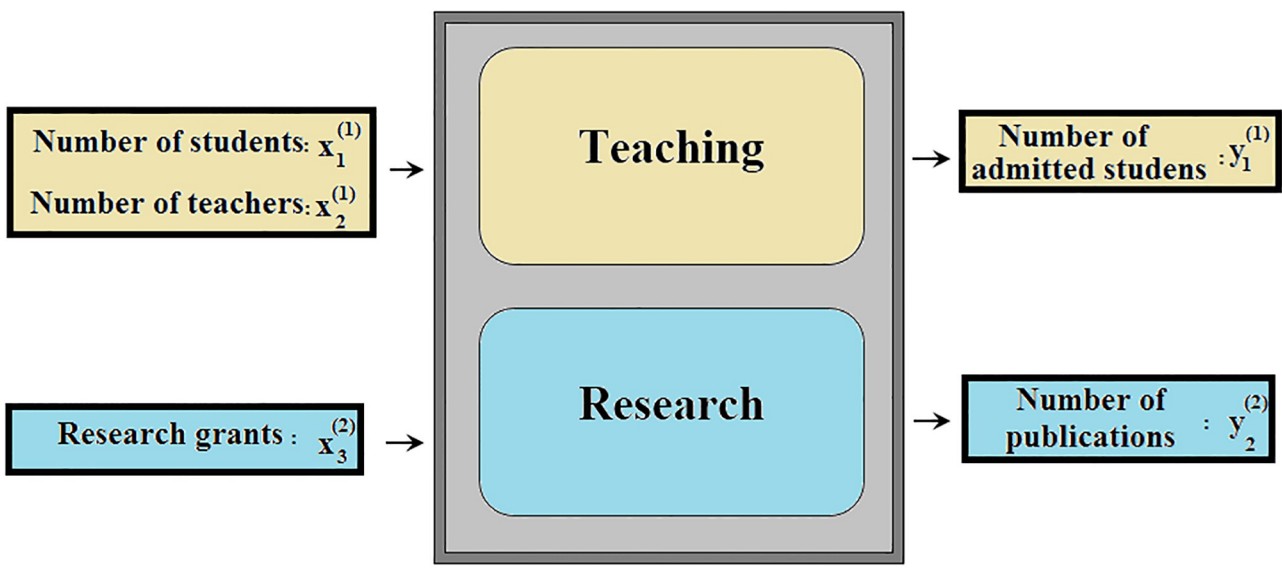

**Fig 3. The structure of the education institution.**

By placing the parameters of Model (21) according to Table 1, this model can be rewritten as Model (22):

$$Max \quad \varphi + \varepsilon(d_1^{+,(1)} + d_2^{+,(2)} - \varepsilon(d_1^{-c,(1)} + d_2^{-c,(1)} + d_3^{-c,(2)}))$$

$$s.t. \quad 100\lambda_1 + 250\lambda_2 + 150\lambda_3 + 180\lambda_4 + 210\lambda_5 + 190\lambda_6 + 230\lambda_7 + 160\lambda_8 + d_1^{-c,(1)} = 100,$$

$$4\lambda_1 + 8\lambda_2 + 5\lambda_3 + 5\lambda_4 + 9\lambda_5 + 6\lambda_6 + 8\lambda_7 + 6\lambda_8 + d_2^{-c,(1)} = 4,$$

$$2550\lambda_1 + 1900\lambda_2 + 3500\lambda_3 + 1950\lambda_4 + 4325\lambda_5 + 3200\lambda_6 + 4000\lambda_7 + 3980\lambda_8 + d_3^{-c,(2)} = 2550,$$

$$90\lambda_1 + 230\lambda_2 + 145\lambda_3 + 140\lambda_4 + 195\lambda_5 + 145\lambda_6 + 200\lambda_7 + 155\lambda_8 - d_1^{+,(1)} = 90\varphi, \quad (22)$$

$$25\lambda_1 + 19\lambda_2 + 16\lambda_3 + 35\lambda_4 + 18\lambda_5 + 25\lambda_6 + 23\lambda_7 + 21\lambda_8 - d_2^{+,(2)} = 25\varphi,$$

$$\lambda_1 + \lambda_2 + \lambda_3 + \lambda_4 + \lambda_5 + \lambda_6 + \lambda_7 + \lambda_8 = 1,$$

$$d_i^{-c,(p)}, d_r^{+,(p)} \geq 0, \quad i = 1,2,3, \quad r = 1,2, \quad p = 1,2,$$

$$\lambda_j \geq 0, j = 1, \ldots, 8.$$

**Table 1. Data of the institutions during the current year.**

| Institution | Teaching Subunits | | | Research Subunits | |
|---|---|---|---|---|---|
| | Number of students | Number of teachers | Number of omitted students | Research grants | Number of publications |
| 1 | 100 | 4 | 90 | 2550 | 25 |
| 2 | 250 | 8 | 230 | 1900 | 19 |
| 3 | 150 | 5 | 145 | 3500 | 16 |
| 4 | 180 | 5 | 140 | 1950 | 35 |
| 5 | 210 | 9 | 195 | 4325 | 18 |
| 6 | 190 | 6 | 145 | 3200 | 25 |
| 7 | 230 | 8 | 200 | 4000 | 23 |
| 8 | 160 | 6 | 155 | 3980 | 21 |

Model (22) is a linear programming problem that can be easily solved in a fraction of a second by using the LP Solver of GAMS software. By solving Model (22), it is concluded that $d_1^{-c,(1)*} = d_2^{-c,(1)*} = d_3^{-c,(2)*} = 0$. This means that Institution 1 exhibits no congestion as a black-box. It is while the research subunit of Institution 1 exhibits congestion comparing to other teaching subunits. It can be seen in Fig 4 that shows the production possibility set of Research Subunit (or the same set $T^{c,(2)}$ as defined in the relation (5)). Complete congestion results corresponding to the institutions as a black-box can be seen in Table 2, individually.

According to Table 2, in addition to Institution1, Institutions 3 and 8 exhibit no congestion in the black-box view while, according to Fig 3, both of them have research subunit including congestion. This issue is the result of ignoring the internal structure of the institutions. It can be addressed using the proposed Model (9). This model to identify the congestion of Institute 'o' ($o \in \{1, 2, \ldots, 8\}$ is as follows:

$$
\begin{aligned}
Max \quad & \frac{1}{2}(\varphi_o^{(1)} + \varphi_o^{(2)}) + \varepsilon\left(d_1^{+,(1)} + d_2^{+,(2)} - \varepsilon(d_1^{-c,(1)} + d_2^{-c,(1)} + d_3^{-c,(2)})\right) \\
s.t. \quad & \sum_{j=1}^{8} \lambda_j^{(1)} x_{ij}^{(1)} + d_i^{-c,(1)} = x_{io}^{(1)}, \quad i = 1, 2, \\
& \sum_{j=1}^{8} \lambda_j^{(2)} x_{3j}^{(2)} + d_3^{-c,(2)} = x_{3o}^{(2)}, \\
& \sum_{j=1}^{8} \lambda_j^{(1)} y_{1j}^{(1)} - d_1^{+,(1)} = \varphi_o^{(1)} y_{11}^{(1)}, \\
& \sum_{j=1}^{8} \lambda_j^{(2)} y_{2j}^{(2)} - d_2^{+,(2)} = \varphi_o^{(2)} y_{21}^{(2)}, \\
& \sum_{j=1}^{n} \lambda_j^{(1)} = 1, \quad \sum_{j=1}^{n} \lambda_j^{(2)} = 1, \\
& d_i^{-c,(p)}, d_r^{+,(p)} \geq 0, \quad i = 1, 2, 3, \ r = 1, 2, \ p = 1, 2, \\
& \lambda_j \geq 0, j = 1, \ldots, 8.
\end{aligned}
\tag{23}
$$

For example Model (23) corresponding to Institution 1 (i.e., $o = 1$) can be writen as Model (24):

$$
\begin{aligned}
Max \quad & \frac{1}{2}(\varphi_1^{(1)} + \varphi_1^{(2)}) + \varepsilon\left(d_1^{+,(1)} + d_2^{+,(2)} - \varepsilon(d_1^{-c,(1)} + d_2^{-c,(1)} + d_3^{-c,(2)})\right) \\
s.t. \quad & 100\lambda_1^{(1)} + 250\lambda_2^{(1)} + 150\lambda_3^{(1)} + 180\lambda_4^{(1)} + 210\lambda_5^{(1)} + 190\lambda_6^{(1)} + 230\lambda_7^{(1)} + 160\lambda_8^{(1)} + d_1^{-c,(1)} = 100, \\
& 4\lambda_1^{(1)} + 8\lambda_2^{(1)} + 5\lambda_3^{(1)} + 5\lambda_4^{(1)} + 9\lambda_5^{(1)} + 6\lambda_6^{(1)} + 8\lambda_7^{(1)} + 6\lambda_8^{(1)} + d_2^{-c,(1)} = 4, \\
& 2550\lambda_1^{(2)} + 1900\lambda_2^{(2)} + 3500\lambda_3^{(2)} + 1950\lambda_4^{(2)} + 4325\lambda_5^{(2)} + \\
& \qquad\qquad\qquad\qquad\qquad 3200\lambda_6^{(2)} + 4000\lambda_7^{(2)} + 3980\lambda_8^{(2)} + d_3^{-c,(2)} = 2550, \\
& 90\lambda_1^{(1)} + 230\lambda_2^{(1)} + 145\lambda_3^{(1)} + 140\lambda_4^{(1)} + 195\lambda_5^{(1)} + 145\lambda_6^{(1)} + 200\lambda_7^{(1)} + 155\lambda_8^{(1)} - d_1^{+,(1)} = 90\varphi_1^{(1)}, \\
& 25\lambda_1^{(2)} + 19\lambda_2^{(2)} + 16\lambda_3^{(2)} + 35\lambda_4^{(2)} + 18\lambda_5^{(2)} + 25\lambda_6^{(2)} + 23\lambda_7^{(2)} + 21\lambda_8^{(2)} - d_2^{+,(2)} = 25\varphi_1^{(2)}, \\
& \lambda_1^{(1)} + \lambda_2^{(1)} + \lambda_3^{(1)} + \lambda_4^{(1)} + \lambda_5^{(1)} + \lambda_6^{(1)} + \lambda_7^{(1)} + \lambda_8^{(1)} = 1, \\
& \lambda_1^{(2)} + \lambda_2^{(2)} + \lambda_3^{(2)} + \lambda_4^{(2)} + \lambda_5^{(2)} + \lambda_6^{(2)} + \lambda_7^{(2)} + \lambda_8^{(2)} = 1, \\
& d_i^{-c,(p)}, d_r^{+,(p)} \geq 0, \quad i = 1, 2, 3, \ r = 1, 2, \ p = 1, 2, \\
& \lambda_j^{(p)} \geq 0, \ , \ p = 1, 2, \ j = 1, \ldots, 8.
\end{aligned}
\tag{24}
$$

By solving Model (24), it is concluded that $\varphi_1^{(1)*} = 1$, $\varphi_1^{(2)*} = 1.4$, $d_1^{+,(1)*} = d_2^{+,(2)*} = 0$, $d_1^{-c,(1)*} = d_2^{-c,(1)*} = 0$ and $d_3^{-c,(2)*} = 600$. This means that the amount of congestion in the research grants of Institute 1 is equal to 600 and no congestion is observed in other inputs.

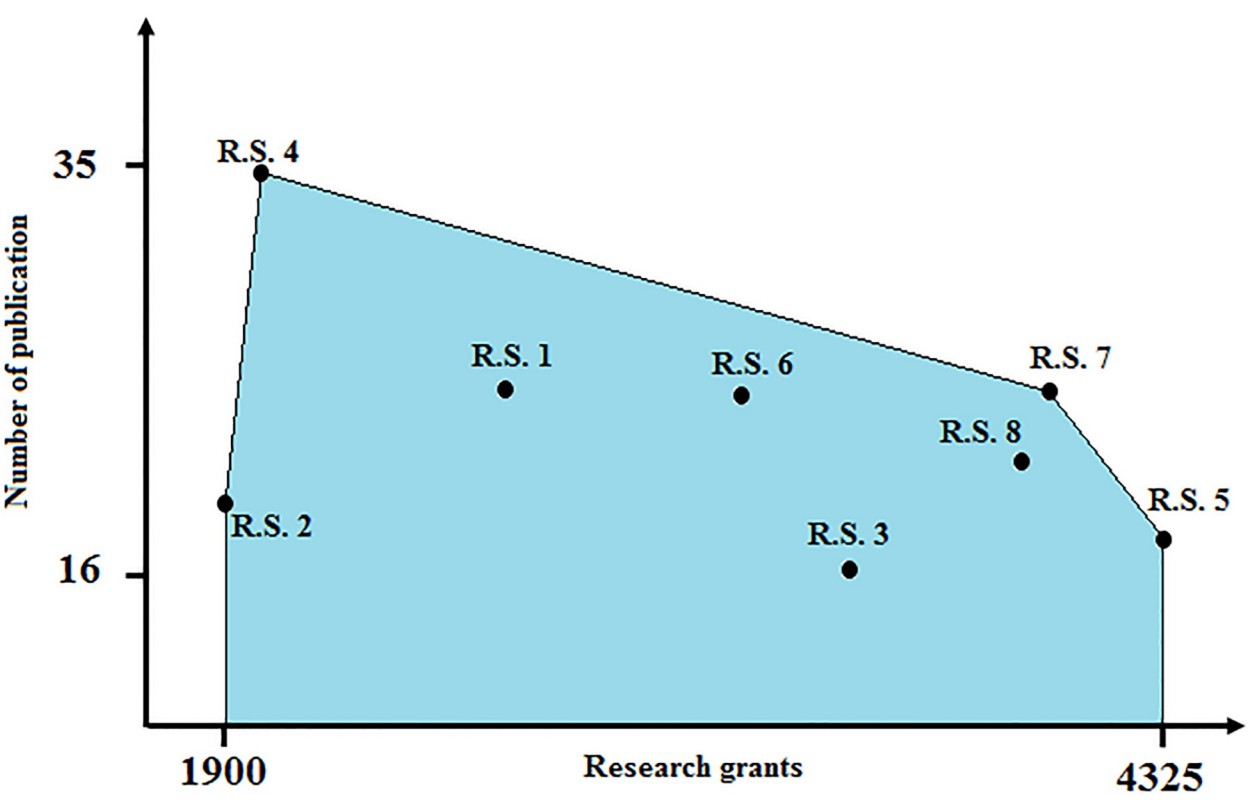

**Fig 4. Production possibility set of research subunit.**

Table 3 represents the results of Model (23) for all $o = 1, 2, \ldots, 8$. As can be seen, the only institution that does not exhibit any congestion is Institution 2.

## 4.2 Case study

The effective use of existing resources is one of the main financial management aims. On the other hand, quantitative analysis is an important component to achieve a correct result. It can be applied by choosing multiple input and output indicators for each organization and then evaluating the organizations according to these indicators. To this end, various

**Table 2. The results of congestion in black-box view.**

| Institution | Congestion amount in black-box view | | |
|:---:|:---:|:---:|:---:|
| | Num. of students | Num. of teachers | R. grants |
| 1 | 0 | 0 | 0 |
| 2 | 0 | 0 | 0 |
| 3 | 0 | 0 | 0 |
| 4 | 0 | 0 | 0 |
| 5 | 0 | 1.89 | 1500.56 |
| 6 | 0 | 0.11 | 756.22 |
| 7 | 0 | 0.85 | 2078.9 |
| 8 | 0 | 0 | 0 |

**Table 3. The results of congestion using the proposed model (23).**

| Institution | Congestion amount using the proposed model | | |
|---|---|---|---|
| | Num. of students | Num. of teachers | R. grants |
| 1 | 0 | 0 | 600 |
| 2 | 0 | 0 | 0 |
| 3 | 0 | 0 | 1550 |
| 4 | 30 | 0 | 0 |
| 5 | 0 | 1.89 | 2375 |
| 6 | 6.67 | 0 | 1250 |
| 7 | 0 | 0.44 | 2050 |
| 8 | 0 | 0 | 2030 |

models and methods with different perspectives can be utilized. Congestion is one of these perspectives whose information can be used as a basis for deciding on organizational adjustment.

Here, the proposed theories and models are applied to analyze the congestion of several Iranian Economic Enterprises in 2018. These Enterprises consist of 3 separate subunits, each of which produces some output by spending its own inputs, as shown in Fig 5: As seen, each Enterprise can be considered as a multi-function parallel system that uses some financial inputs to produce financial outputs in 3 separate subunits. The data set corresponding to these Enterprises is listed in Table 4. Personnel privilege is an indicator calculated based on components such as the number of employees, their background, or age. The other input/output data are in terms of 10000 IRR (The Iranian Rial (ISO 4217 code IRR) is the currency of Iran). This section aims to compare the results of the proposed method with the results of the existing traditional single-model (2) that treats these enterprises like a black-box. The single-model (2) to evaluate the congestion of *o-th* Enterprise is as Model (25):

$$
\begin{aligned}
Max \quad & \varphi + \varepsilon \left( \sum_{r=1}^{3} s_r^+ - \varepsilon \sum_{i=1}^{6} s_i^{-c} \right) \\
s.t. \quad & \sum_{j=1}^{51} \lambda_j x_{ij} + s_i^{-c} = x_{io} \qquad i = 1, ..., 6, \\
& \sum_{j=1}^{51} \lambda_j y_{rj} - s_r^+ = \varphi y_{ro} \qquad r = 1, 2, 3, \\
& \sum_{j=1}^{51} \lambda_j = 1, \quad \lambda_j \geq 0 \qquad j = 1, ..., 51, \\
& s_i^{-c}, s_r^+ \geq 0 \qquad i = 1, ..., 6 , \; r = 1, 2, 3.
\end{aligned}
\tag{25}
$$

where, the corresponding notation are listed in Table 5:

Table 6 represents the obtained results from Model (25) so that the congestion values in the input components are listed in columns 6 to 11. Moreover, the last column shows the presence or absence of congestion in the black-box view. On the other hand, the proposed model (9) to evaluate the congestion of *o*-th Enterprise is as Model (26), where, the related notation can be

**Fig 5. Economic enterprises consist of 3 separate subunits.**

seen in Table 7.

$$Max \quad \frac{1}{3}\left(\varphi_o^{(1)} + \varphi_o^{(2)} + \varphi_o^{(3)}\right) + \varepsilon(d_1^{+,(1)} + d_2^{+,(2)} + d_3^{+,(3)} -$$

$$\varepsilon(d_1^{-c,(1)} + d_2^{-c,(1)} + d_3^{-c,(2)} + d_4^{-c,(3)} + d_5^{-c,(3)} + d_6^{-c,(3)})\Big)$$

$$s.t. \quad \sum_{j=1}^{51} \lambda_j^{(1)} x_{ij}^{(1)} + d_i^{-c,(1)} = x_{io}^{(1)}, \quad i = 1, 2,$$

$$\sum_{j=1}^{51} \lambda_j^{(2)} x_{3j}^{(2)} + d_3^{-c,(2)} = x_{3o}^{(2)},$$

$$\sum_{j=1}^{51} \lambda_j^{(3)} x_{ij}^{(3)} + d_i^{-c,(3)} = x_{io}^{(3)}, \quad i = 4, 5, 6,$$

$$\sum_{j=1}^{51} \lambda_j^{(1)} y_{1j}^{(1)} - d_1^{+,(1)} = \varphi_o^{(1)} y_{1o}^{(1)},$$

$$\sum_{j=1}^{51} \lambda_j^{(2)} y_{2j}^{(2)} - d_2^{+,(2)} = \varphi_o^{(2)} y_{2o}^{(2)}, \qquad (26)$$

$$\sum_{j=1}^{51} \lambda_j^{(3)} y_{3j}^{(3)} - d_3^{+,(3)} = \varphi_o^{(3)} y_{3o}^{(3)},$$

$$\sum_{j=1}^{51} \lambda_j^{(1)} = 1, \quad \sum_{j=1}^{51} \lambda_j^{(2)} = 1, \quad \sum_{j=1}^{51} \lambda_j^{(3)} = 1$$

$$d_i^{-c,(p)}, d_r^{+,(p)} \geq 0, \quad i = 1, .., 6, \quad r = 1, 2, 3, \quad p = 1, 2, 3$$

$$\lambda_j \geq 0, j = 1, \ldots, 51.$$

The results of Model (26) are represented in Table 8. Columns 8 to 10 indicate the amount of congestion in the subunits Profitability, Service, and Job creating, respectively. The noteworthy point about Table 8 is that all slack variables $d_r^{+,(k)} (k = 1, ..., q, r = 1, ..., s)$ are zero. It is because there is only one output in all three subunits, and, in this case, $\varphi_o^{(k)}$s $(k = 1, \ldots, q)$ variables are alone sufficient to achieve the maximum possible increase in the output components. Another point is that the units that exhibit congestion according to Model (25), also exhibit congestion by using Model (26). In addition, several other units exhibit congestion that

**Table 4. The data for 51 Iranian Economic Enterprises in 2018.**

| Enterprise | Profitability | | | Service | | | Job creating | | |
|---|---|---|---|---|---|---|---|---|---|
| | Pers. privilege | paid profit | Rec. profit | Pers. privilege | Rec. commission | Pers. privilege | Total deposits | Other resources | G. facilities |
| 1 | 1.98 | 570344 | 385360 | 23.94 | 592650 | 26.32 | 4498178 | 2383092 | 10652033 |
| 2 | 18.65 | 170704 | 6039273 | 0.11 | 28512 | 3.5 | 1866264 | 315 | 4228374 |
| 3 | 0.15 | 151329 | 1644267 | 4.81 | 257987 | 37.77 | 1553664 | 679500 | 7380250 |
| 4 | 4.7 | 567733 | 3608075 | 20.91 | 401674 | 25.92 | 4947282 | 2586671 | 17159727 |
| 5 | 19.77 | 157864 | 3518930 | 27.65 | 115735 | 26.21 | 1161030 | 382000 | 3122199 |
| 6 | 28.43 | 124776 | 387206 | 22.33 | 14336 | 24.95 | 1519785 | 22688 | 2199970 |
| 7 | 39.84 | 1079358 | 9064527 | 3.30 | 727242 | 10.67 | 9172340 | 1753854 | 22046946 |
| 8 | 27.48 | 747457 | 14582505 | 8.98 | 377384 | 12.01 | 5645452 | 1918697 | 10133369 |
| 9 | 33.3 | 311867 | 872759 | 7.22 | 266907 | 0.28 | 2351810 | 1665450 | 3174696 |
| 10 | 12.05 | 1031120 | 9470801 | 8.17 | 472582 | 0.25 | 7242326 | 9398647 | 15869047 |
| 11 | 1.61 | 209183 | 2237761 | 12.74 | 291361 | 4.32 | 1525359 | 42900 | 4484253 |
| 12 | 18.11 | 386350 | 4130470 | 28.89 | 151206 | 34.48 | 4140500 | 526 | 6665945 |
| 13 | 36.19 | 378308 | 2882747 | 26.49 | 520003 | 19.46 | 3026972 | 119348 | 5060971 |
| 14 | 12.75 | 730378 | 2560681 | 30.45 | 845250 | 39.61 | 7076859 | 1402699 | 8322555 |
| 15 | 28.66 | 213341 | 1669132 | 28.63 | 135468 | 7.3 | 2022618 | 515000 | 3448174 |
| 16 | 17.54 | 250836 | 1905944 | 35.14 | 175338 | 29.99 | 2186122 | 329332 | 2911501 |
| 17 | 17.85 | 285197 | 366536 | 15.90 | 257957 | 27.59 | 2618874 | 1565260 | 2285021 |
| 18 | 23.72 | 2274755 | 48010182 | 5.71 | 550407 | 23.56 | 12890550 | 3950398 | 34781183 |
| 19 | 8.31 | 502885 | 4388860 | 39.90 | 740832 | 23.5 | 3724091 | 3239522 | 6041657 |
| 20 | 0.84 | 786803 | 2804317 | 23.30 | 1835979 | 34.81 | 6505526 | 9568680 | 8701973 |
| 21 | 21.97 | 464246 | 1673163 | 18.03 | 546425 | 32.08 | 3377997 | 2165000 | 4291442 |
| 22 | 35.97 | 888387 | 1109861 | 33.49 | 779096 | 18.83 | 6534041 | 186518 | 3818066 |
| 23 | 10.41 | 5695 | 2525752 | 12.81 | 542137 | 18.53 | 4247877 | 1580832 | 3423831 |
| 24 | 20.38 | 1050066 | 4111140 | 12.24 | 1032304 | 7.36 | 7157137 | 3617596 | 8700439 |
| 25 | 23.28 | 647568 | 6678399 | 24.40 | 307126 | 37.05 | 5421327 | 224444 | 7943473 |
| 26 | 8.65 | 732311 | 1678358 | 23.17 | 311063 | 9.29 | 5012300 | 112000 | 6792835 |
| 27 | 3.04 | 503878 | 1825169 | 11.26 | 715890 | 14.94 | 3963067 | 3122156 | 12005850 |
| 28 | 27.99 | 192116 | 2477375 | 17.20 | 108808 | 12.29 | 1676823 | 51748 | 2101435 |
| 29 | 7.16 | 215974 | 470122 | 31.48 | 141890 | 35.48 | 1547215 | 486325 | 1542516 |
| 30 | 10.34 | 422507 | 1431894 | 14.18 | 584696 | 35.66 | 3500632 | 3479025 | 7737054 |
| 31 | 32.55 | 201180 | 1021176 | 27.84 | 128748 | 4.28 | 1778190 | 56286 | 1432375 |
| 32 | 21.61 | 267681 | 1230230 | 16.12 | 190067 | 13.6 | 1934511 | 2362193 | 1376649 |
| 33 | 4.52 | 240327 | 659532 | 10.48 | 555254 | 32.52 | 3417999 | 191400 | 2263374 |
| 34 | 36.38 | 539621 | 930794 | 16.51 | 1110018 | 4.62 | 4767901 | 7584114 | 5786974 |
| 35 | 35.83 | 258695 | 301722 | 30.82 | 445648 | 31.04 | 1843651 | 2128060 | 1926108 |
| 36 | 11.1 | 287351 | 236406 | 5.10 | 139391 | 5.61 | 2444899 | 5263 | 1423588 |
| 37 | 22.14 | 324327 | 1009391 | 20.65 | 486688 | 6.62 | 2544487 | 757938 | 3139247 |
| 38 | 21.15 | 1151127 | 3104915 | 38.09 | 3811906 | 30.08 | 9947533 | 16242839 | 10237431 |
| 39 | 18.72 | 303012 | 3455449 | 6.97 | 178901 | 19.24 | 2274082 | 50000 | 2161360 |
| 40 | 26.92 | 513373 | 838165 | 34.35 | 256331 | 22.02 | 4547810 | 150000 | 1865312 |
| 41 | 17.03 | 951481 | 3307344 | 36.87 | 975447 | 34.05 | 7567546 | 10235468 | 8541061 |
| 42 | 10.28 | 3080 | 993266 | 9.88 | 162474 | 6.23 | 2667901 | 100000 | 2027599 |
| 43 | 25.28 | 204132 | 1137743 | 26.25 | 49696 | 8.73 | 1542902 | 1200 | 848146 |
| 44 | 0.52 | 50371 | 314533 | 28.73 | 136400 | 3.1 | 1675934 | 257 | 1656344 |
| 45 | 0.94 | 1694107 | 4053550 | 19.13 | 913782 | 24.9 | 13555750 | 3379699 | 16176051 |
| 46 | 29.55 | 190578 | 739454 | 0.99 | 76809 | 25.43 | 1963070 | 6263 | 2122453 |

*(Continued)*

**Table 4.** (Continued)

| Enterprise | Profitability | | | Service | | | Job creating | | |
|---|---|---|---|---|---|---|---|---|---|
| | Pers. privilege | paid profit | Rec. profit | Pers. privilege | Rec. commission | Pers. privilege | Total deposits | Other resources | G. facilities |
| 47 | 35.13 | 431866 | 1592638 | 31.33 | 491494 | 35.96 | 3586788 | 1568484 | 3656533 |
| 48 | 22.64 | 282195 | 2014254 | 23.70 | 258896 | 10.52 | 2424545 | 10000 | 2705216 |
| 49 | 33.99 | 213896 | 2501476 | 30.91 | 95037 | 31.5 | 2003669 | 50000 | 3039289 |
| 50 | 0.53 | 237636 | 1573939 | 27.34 | 61907 | 38.43 | 2732013 | 444000 | 3078221 |
| 51 | 28.12 | 224872 | 2423595 | 8.87 | 294191 | 24.41 | 1808102 | 92860 | 2667443 |

previously did not exhibit congestion according to Model (25). It indicates more comprehensiveness and flexibility of the proposed method than the black-box approach.

Here, to compare the performance and results of the proposed method with the existing method as the black-box approach, it is better to take a closer look at one of the enterprises. For this purpose, consider Enterprise 51. The input values of this enterprise in the first subunit are $x_{1,51}^{(1)} = 28.12$ and $x_{2,51}^{(1)} = 224872$; in the second subunit, it is equal to $x_{3,51}^{(2)} = 8.87$; and, in the third subunit, they are equal to $x_{4,51}^{(3)} = 24.41$, $x_{5,51}^{(3)} = 1808102$, and, $x_{6,51}^{(3)} = 92860$. If we pay attention to the data in Table 4, the input amount of this enterprise in the second subsection is one of the lowest values compared to the amount of the same input in other institutions. This means that Enterprise 51 probably does not exhibit congestion in terms of the second subunit which includes this input component. But the same issue in the black-box view, when the congestion of institutions is examined by considering the input components of all subunits together, can affect the existence of congestion in other subunits and prevent the exhibiting the congestion throughout the system. This is evident in the results of Model (25), in which, Enterprise 51 is identified without any congestion. However, in Model (26), where the subunits separately are examined, the same enterprise exhibits congestion. With careful consideration of the data in Table 8, as expected, no congestion is seen in the second subunit of institute 51, but, the presence of congestion in the first and third subunits has led to exhibit the congestion in the entire institution. That is the difference between a black-box view and the proposed approach.

## 5 Comparison of the proposed method with the existing view as a black-box

First of all, it is worth noting that the proposed method is the same as the single-model (2) with a different approach. The proposed method uses the same single-model (2) to detect and

**Table 5. The notation corresponding to model (25).**

| Subunit | Inputs | Outputs |
|---|---|---|
| Profitability | Personnel privilege: $x_{1j}$ | Received profit: $y_{1j}$ |
| | paid profit: $x_{2j}$ | |
| Service | Personnel privilege: $x_{3j}$ | Received commission: $y_{2j}$ |
| Job creating | Personnel privilege: $x_{4j}$ | Granting facilities: $y_{3j}$ |
| | Total deposits: $x_{5j}$ | |
| | Other resources: $x_{6j}$ | |

**Table 6. The result of congestion in the black-box view.**

| Enterprise | $\varphi^*$ | $s_r^{+*}$ | | | $s_i^{-c*}$ | | | | | | Exhibiting cong. |
|---|---|---|---|---|---|---|---|---|---|---|---|
| | | $s_1^{+*}$ | $s_2^{+*}$ | $s_3^{+*}$ | $s_1^{-c*}$ | $s_2^{-c*}$ | $s_3^{-c*}$ | $s_4^{-c*}$ | $s_5^{-c*}$ | $s_6^{-c*}$ | |
| 1 | 1.03 | 1708539.58 | 0 | 0 | 0 | 13708.66 | 13.13 | 4.73 | 0 | 0 | Yes |
| 2 | 1 | 0 | 0 | 0 | 0 | 0 | 0 | 0 | 0 | 0 | No |
| 3 | 1 | 0 | 0 | 0 | 0 | 0 | 0 | 0 | 0 | 0 | No |
| 4 | 1 | 0 | 0 | 0 | 0 | 0 | 0 | 0 | 0 | 0 | No |
| 5 | 1 | 0 | 0 | 0 | 0 | 0 | 0 | 0 | 0 | 0 | No |
| 6 | 1 | 0 | 0 | 0 | 0 | 0 | 0 | 0 | 0 | 0 | No |
| 7 | 1 | 0 | 0 | 0 | 0 | 0 | 0 | 0 | 0 | 0 | No |
| 8 | 1.11 | 0 | 0 | 1256063.99 | 10.34 | 0 | 2.77 | 0 | 253158.61 | 0 | Yes |
| 9 | 1 | 0 | 0 | 0 | 0 | 0 | 0 | 0 | 0 | 0 | No |
| 10 | 1 | 0 | 0 | 0 | 0 | 0 | 0 | 0 | 0 | 0 | No |
| 11 | 1 | 0 | 0 | 0 | 0 | 0 | 0 | 0 | 0 | 0 | No |
| 12 | 1 | 0 | 0 | 0 | 0 | 0 | 0 | 0 | 0 | 0 | No |
| 13 | 1 | 0 | 0 | 0 | 0 | 0 | 0 | 0 | 0 | 0 | No |
| 14 | 1 | 0 | 0 | 0 | 0 | 0 | 0 | 0 | 0 | 0 | No |
| 15 | 1.59 | 303156.75 | 0 | 0 | 22.9 | 0 | 15.51 | 0 | 0 | 181131.25 | Yes |
| 16 | 1.86 | 0 | 0 | 0 | 11.09 | 0 | 22.52 | 19.95 | 0 | 0 | Yes |
| 17 | 2.41 | 1048209.04 | 0 | 0 | 12.03 | 0 | 2.81 | 5.62 | 0 | 0 | Yes |
| 18 | 1 | 0 | 0 | 0 | 0 | 0 | 0 | 0 | 0 | 0 | No |
| 19 | 1.24 | 0 | 0 | 0 | 0.2 | 548.75 | 22.11 | 9.71 | 0 | 0 | Yes |
| 20 | 1 | 0 | 0 | 0 | 0 | 0 | 0 | 0 | 0 | 0 | No |
| 21 | 1.47 | 386421.35 | 0 | 0 | 8.55 | 51054.68 | 0 | 20.45 | 0 | 0 | Yes |
| 22 | 1 | 0 | 0 | 0 | 0 | 0 | 0 | 0 | 0 | 0 | No |
| 23 | 1 | 0 | 0 | 0 | 0 | 0 | 0 | 0 | 0 | 0 | No |
| 24 | 1 | 0 | 0 | 0 | 0 | 0 | 0 | 0 | 0 | 0 | No |
| 25 | 1 | 0 | 0 | 0 | 0 | 0 | 0 | 0 | 0 | 0 | No |
| 26 | 1 | 0 | 0 | 0 | 0 | 0 | 0 | 0 | 0 | 0 | No |
| 27 | 1 | 0 | 0 | 0 | 0 | 0 | 0 | 0 | 0 | 0 | No |
| 28 | 1.6 | 0 | 0 | 982626.01 | 18.31 | 0 | 9.59 | 7.68 | 0 | 0 | Yes |
| 29 | 2.13 | 1051987.8 | 0 | 785894.79 | 0 | 3092.3 | 15.51 | 26.4 | 0 | 148082.53 | Yes |
| 30 | 1.31 | 0 | 0 | 0 | 7.02 | 4930.2 | 3.22 | 10.56 | 0 | 316032.98 | Yes |
| 31 | 2.19 | 0 | 0 | 1215328.31 | 30.55 | 0 | 14.65 | 0 | 225131.26 | 0 | Yes |
| 32 | 2.27 | 0 | 0 | 1398691.5 | 14.31 | 19933.97 | 0 | 3.74 | 0 | 1379973.94 | Yes |
| 33 | 1 | 0 | 0 | 0 | 0 | 0 | 0 | 0 | 0 | 0 | No |
| 34 | 1 | 0 | 0 | 0 | 0 | 0 | 0 | 0 | 0 | 0 | No |
| 35 | 1 | 0 | 0 | 0 | 0 | 0 | 0 | 0 | 0 | 0 | No |
| 36 | 1 | 0 | 0 | 0 | 0 | 0 | 0 | 0 | 0 | 0 | No |
| 37 | 1 | 0 | 0 | 0 | 0 | 0 | 0 | 0 | 0 | 0 | Yes |
| 38 | 1 | 0 | 0 | 0 | 0 | 0 | 0 | 0 | 0 | 0 | No |
| 39 | 1.2 | 0 | 0 | 1447993.35 | 3.84 | 73103.18 | 0 | 10.01 | 0 | 0 | Yes |
| 40 | 2.28 | 0 | 0 | 0 | 0 | 0 | 9.72 | 3.04 | 443159.59 | 0 | Yes |
| 41 | 1.84 | 675179.74 | 0 | 0 | 4.25 | 19243.17 | 10.16 | 6.63 | 0 | 2024437.82 | Yes |
| 42 | 1 | 0 | 0 | 0 | 0 | 0 | 0 | 0 | 0 | 0 | No |
| 43 | 1 | 0 | 0 | 0 | 0 | 0 | 0 | 0 | 0 | 0 | No |
| 44 | 1 | 0 | 0 | 0 | 0 | 0 | 0 | 0 | 0 | 0 | No |
| 45 | 1 | 0 | 0 | 0 | 0 | 0 | 0 | 0 | 0 | 0 | No |
| 46 | 1 | 0 | 0 | 0 | 0 | 0 | 0 | 0 | 0 | 0 | No |

(*Continued*)

**Table 6.** (Continued)

| Enterprise | $\varphi^*$ | $s_r^{+*}$ | | | $s_i^{-c*}$ | | | | | | Exhibiting cong. |
|---|---|---|---|---|---|---|---|---|---|---|---|
| | | $s_1^{+*}$ | $s_2^{+*}$ | $s_3^{+*}$ | $s_1^{-c*}$ | $s_2^{-c*}$ | $s_3^{-c*}$ | $s_4^{-c*}$ | $s_5^{-c*}$ | $s_6^{-c*}$ | |
| 47 | 1.6 | 331035.79 | 0 | 0 | 3.05 | 0 | 5.8 | 14.36 | 0 | 0 | Yes |
| 48 | 1 | 0 | 0 | 0 | 0 | 0 | 0 | 0 | 0 | 0 | No |
| 49 | 1.58 | 42122.89 | 15287.62 | 0 | 23.42 | 0 | 21.55 | 22.03 | 0 | 0 | Yes |
| 50 | 1.02 | 0 | 186242.87 | 2826212.64 | 0 | 86833.51 | 17.51 | 12.9 | 1168268.38 | 0 | Yes |
| 51 | 1 | 0 | 0 | 0 | 0 | 0 | 0 | 0 | 0 | 0 | No |

determine the congestion, except that each subunit is examined separately. In this way, the computational volume of the model is reduced and on the other hand, more information about the input and output components is obtained. In general, to evaluate the congestion, there are several important advantages of the proposed method compared to the black-box view in dealing with multi-function parallel systems. Consider $DMU_o$ with the $q$ parallel subunits, as shown in Fig 2. Then, the following points are always valid:

- To evaluate the congestion of $DMU_o$ by using the proposed method, the single-model (2) should be solved $q$ times (number of the existing subunits). Here, although the number of solving times is more than the black-box view, the computational volume will be much less. In fact, in the black-box view, Single-model (2) has $(m + s + 1)$ constraints along with the $(n + m + s)$ variables, while in the proposed method, this model has $((m^{(k)} - m^{(k-1)}) + (s^{(k)} - s^{(k-1)}) + 1)$ constraints with the $(n + (m^{(k)} - m^{(k-1)}) + (s^{(k)} - s^{(k-1)}))$ variables, in evaluating the congestion of the $k$-th Subunit ($m^{(0)} = s^{(0)} = 0$ and $m^{(q)} = m$, $s^{(q)} = s$). On the other hand, the most important factor in solving a linear programming model is the size of its basis, not the number of times it is solved. Therefore, utilizing the proposed method to evaluate the congestion of the multi-function parallel systems is economical in comparison with the black-box view from a computational viewpoint.

- According to Theorem 3.4, the next advantage is the greater flexibility of the proposed method in obtaining the congestion information. In better words, in addition to identifying units that have congestion in the black-box view, Model (9) can also get more information about the congestion. That is seen in the examples of the previous section.

- By using the proposed approach, the performance of each subunit separately can be evaluated and the appropriate decision can be made for the subunits. In this way, each subunit can serve the whole unit independently and thus increase the ability of the system. Of course,

**Table 7. The notation corresponding to model (26).**

| Subunit | Inputs | Outputs | Congestion |
|---|---|---|---|
| Profitability | Personnel privilege: $x_{1j}^{(1)}$ | Received profit: $y_{1o}^{(1)}$ | Congestion of Pers. privilege: $d_1^{-c,(1)}$ |
| | paid profit: $x_{2j}^{(1)}$ | | Congestion of paid profit: $d_2^{-c,(1)}$ |
| Service | Personnel privilege: $x_{3j}^{(2)}$ | Received commission: $y_{2o}^{(2)}$ | Congestion of Pers. privilege: $d_3^{-c,(2)}$ |
| Job creating | Personnel privilege: $x_{4j}^{(3)}$ | Granting facilities: $y_{3o}^{(3)}$ | Congestion of Pers. privilege: $d_4^{-c,(3)}$ |
| | Total deposits:$x_{5j}^{(3)}$ | | Congestion of Total deposits: $d_5^{-c,(3)}$ |
| | Other resources:$x_{6j}^{(3)}$ | | Congestion of Pers. Other resources: $d_6^{-c,(3)}$ |

**Table 8. The result of congestion corresponding to the proposed model.**

| Enterprise | $\varphi_o^{(k)*}$ | | | $d_r^{+,(p)}$ | | | $d_i^{-c,(p)}$ | | | | | | Exhibiting cong. |
|---|---|---|---|---|---|---|---|---|---|---|---|---|---|
| | $\varphi_o^{(1)*}$ | $\varphi_o^{(2)*}$ | $\varphi_o^{(3)*}$ | $d_1^{+,(1)*}$ | $d_2^{+,(2)*}$ | $d_3^{+,(3)*}$ | $d_1^{-c,(1)*}$ | $d_2^{-c,(1)*}$ | $d_3^{-c,(2)*}$ | $d_4^{-c,(3)*}$ | $d_5^{-c,(3)*}$ | $d_6^{-c,(3)*}$ | |
| 1 | 13.99 | 4.32 | 1.48 | 0 | 0 | 0 | 0 | 0 | 0 | 0 | 0 | 70424.98 | Yes |
| 2 | 1 | 1 | 1 | 0 | 0 | 0 | 0 | 0 | 0 | 0 | 0 | 0 | No |
| 3 | 1 | 3.34 | 1 | 0 | 0 | 0 | 0 | 0 | 0 | 0 | 0 | 0 | No |
| 4 | 2.94 | 5.7 | 1 | 0 | 0 | 0 | 0 | 0 | 0 | 0 | 0 | 0 | No |
| 5 | 1.64 | 24.94 | 1 | 0 | 0 | 0 | 1.76 | 0 | 0 | 0 | 0 | 0 | Yes |
| 6 | 13.07 | 168.43 | 1 | 0 | 0 | 0 | 12.07 | 0 | 0 | 0 | 0 | 0 | Yes |
| 7 | 2.67 | 1 | 1 | 0 | 0 | 0 | 19 | 0 | 0 | 0 | 0 | 0 | Yes |
| 8 | 1.2 | 3.26 | 1.54 | 0 | 0 | 0 | 7.44 | 0 | 0 | 0 | 0 | 397398.39 | Yes |
| 9 | 10.15 | 4.03 | 1 | 0 | 0 | 0 | 14.31 | 0 | 0 | 0 | 0 | 0 | Yes |
| 10 | 2.3 | 2.45 | 1 | 0 | 0 | 0 | 0 | 0 | 0 | 0 | 0 | 0 | No |
| 11 | 1.51 | 5.37 | 1 | 0 | 0 | 0 | 0 | 0 | 0 | 0 | 0 | 0 | No |
| 12 | 2.5 | 19.82 | 1 | 0 | 0 | 0 | 0 | 0 | 0 | 0 | 0 | 0 | No |
| 13 | 3.53 | 5.35 | 1.25 | 0 | 0 | 0 | 17.04 | 0 | 0 | 2.32 | 0 | 0 | Yes |
| 14 | 6.43 | 3.71 | 2.13 | 0 | 0 | 0 | 0 | 0 | 0 | 24.44 | 0 | 0 | Yes |
| 15 | 4.13 | 21.95 | 1.82 | 0 | 0 | 0 | 9.91 | 0 | 0 | 0 | 0 | 116187.6 | Yes |
| 16 | 3.99 | 20.25 | 2.28 | 0 | 0 | 0 | 0 | 0 | 0 | 17.77 | 0 | 0 | Yes |
| 17 | 22.62 | 7.15 | 4.34 | 0 | 0 | 0 | 0 | 0 | 0 | 0 | 0 | 406651.15 | Yes |
| 18 | 1 | 1.71 | 1 | 0 | 0 | 0 | 0 | 0 | 0 | 0 | 0 | 0 | No |
| 19 | 2.5 | 5.15 | 2.16 | 0 | 0 | 0 | 0 | 0 | 1.81 | 0 | 0 | 1464064.12 | Yes |
| 20 | 1.19 | 1.36 | 2.37 | 0 | 0 | 0 | 0 | 0 | 0 | 9.35 | 0 | 6714484.42 | Yes |
| 21 | 7.11 | 3.72 | 2.94 | 0 | 0 | 0 | 2.61 | 0 | 0 | 0.68 | 0 | 460247.6 | Yes |
| 22 | 18.34 | 4.37 | 2.05 | 0 | 0 | 0 | 15.59 | 0 | 0 | 0 | 1553845.15 | 0 | Yes |
| 23 | 1 | 2.9 | 3.92 | 0 | 0 | 0 | 0 | 0 | 0 | 0 | 0 | 0 | No |
| 24 | 5.73 | 1.47 | 1.99 | 0 | 0 | 0 | 0 | 0 | 0 | 0 | 0 | 0 | No |
| 25 | 2.33 | 8.46 | 1.09 | 0 | 0 | 0 | 3.48 | 0 | 0 | 5.61 | 638209.33 | 0 | Yes |
| 26 | 9.09 | 8 | 1 | 0 | 0 | 0 | 0 | 0 | 0 | 0 | 0 | 0 | No |
| 27 | 4.05 | 2 | 1.04 | 0 | 0 | 0 | 0 | 0 | 0 | 0 | 0 | 1679493.1 | Yes |
| 28 | 2.61 | 18.01 | 2.23 | 0 | 0 | 0 | 9.29 | 0 | 0 | 6.73 | 0 | 0 | Yes |
| 29 | 11.42 | 22.73 | 4.22 | 0 | 0 | 0 | 0 | 0 | 0 | 7.88 | 0 | 0 | Yes |
| 30 | 7.06 | 2.89 | 1.68 | 0 | 0 | 0 | 0 | 0 | 0 | 4.69 | 0 | 1705353.26 | Yes |
| 31 | 6.51 | 22.55 | 3.28 | 0 | 0 | 0 | 13.83 | 0 | 0 | 0 | 0 | 0 | Yes |
| 32 | 6.48 | 9.81 | 4.77 | 0 | 0 | 0 | 2.73 | 0 | 0 | 0 | 0 | 1891231.19 | Yes |
| 33 | 7.59 | 2.46 | 3.16 | 0 | 0 | 0 | 0 | 0 | 0 | 14.36 | 0 | 0 | Yes |
| 34 | 14.39 | 1.71 | 2.02 | 0 | 0 | 0 | 16.84 | 0 | 0 | 0 | 0 | 3636958.34 | Yes |
| 35 | 25.83 | 7.11 | 4.02 | 0 | 0 | 0 | 16.97 | 0 | 0 | 0 | 0 | 1391369.72 | Yes |
| 36 | 33.32 | 6.36 | 3.15 | 0 | 0 | 0 | 0 | 0 | 0 | 0 | 303499.32 | 0 | Yes |
| 37 | 9.02 | 4.66 | 2.34 | 0 | 0 | 0 | 3.12 | 0 | 0 | 0 | 0 | 314936.11 | Yes |
| 38 | 8.24 | 1 | 2.76 | 0 | 0 | 0 | 0.14 | 0 | 0 | 5.65 | 0 | 12797708.07 | Yes |
| 39 | 2.51 | 5.88 | 2.42 | 0 | 0 | 0 | 0 | 0 | 0 | 7.14 | 0 | 0 | Yes |
| 40 | 15.36 | 13.58 | 4.01 | 0 | 0 | 0 | 7.44 | 0 | 0 | 0 | 0 | 0 | Yes |
| 41 | 6.51 | 3.8 | 2.69 | 0 | 0 | 0 | 0 | 0 | 0 | 8.91 | 0 | 7198941.25 | Yes |
| 42 | 1 | 8.07 | 2.73 | 0 | 0 | 0 | 0 | 0 | 0 | 0 | 0 | 0 | No |
| 43 | 5.89 | 55.58 | 1 | 0 | 0 | 0 | 6.55 | 0 | 0 | 0 | 0 | 0 | Yes |
| 44 | 1 | 21.86 | 1 | 0 | 0 | 0 | 0 | 0 | 0 | 0 | 0 | 0 | No |
| 45 | 1 | 2.33 | 1.95 | 0 | 0 | 0 | 0 | 0 | 0 | 4.69 | 1631253.37 | 0 | Yes |
| 46 | 8.7 | 2.88 | 2.08 | 0 | 0 | 0 | 10.85 | 0 | 0 | 19.85 | 0 | 0 | Yes |

(*Continued*)

**Table 8.** (Continued)

| Enterprise | $\varphi_o^{(k)*}$ | | | $d_r^{+,(p)}$ | | | $d_i^{-c,(p)}$ | | | | | | Exhibiting cong. |
|---|---|---|---|---|---|---|---|---|---|---|---|---|---|
| | $\varphi_o^{(1)*}$ | $\varphi_o^{(2)*}$ | $\varphi_o^{(3)*}$ | $d_1^{+,(1)*}$ | $d_2^{+,(2)*}$ | $d_3^{+,(3)*}$ | $d_1^{-c,(1)*}$ | $d_2^{-c,(1)*}$ | $d_3^{-c,(2)*}$ | $d_4^{-c,(3)*}$ | $d_5^{-c,(3)*}$ | $d_6^{-c,(3)*}$ | |
| 47 | 7.06 | 6.54 | 3.5 | 0 | 0 | 0 | 15.85 | 0 | 0 | 3.21 | 0 | 0 | Yes |
| 48 | 4.1 | 9.8 | 1.8 | 0 | 0 | 0 | 3.72 | 0 | 0 | 0 | 0 | 0 | Yes |
| 49 | 2.76 | 33.41 | 1.64 | 0 | 0 | 0 | 15.24 | 0 | 0 | 22.33 | 0 | 0 | Yes |
| 50 | 1.54 | 46.18 | 2.55 | 0 | 0 | 0 | 0 | 0 | 0 | 26.15 | 0 | 0 | Yes |
| 51 | 2.94 | 4.15 | 1.89 | 0 | 0 | 0 | 9.34 | 0 | 0 | 19.26 | 0 | 0 | Yes |

although the subunits are considered separately, the system manager decides on changes following the general policies of the system. In other words, the mere presence of congestion in an input component can not be a reason for a definite reduction of this component.

Anyway, according to the above, it should be noted that the efficiency of the proposed method is better determined when the number of subunits increases.

## 6 Conclusions

As known, recognizing the congestion status of DMUs is one of the most significant topics in the DEA literature. Nonetheless, there is very limited literature available based on the congestion of the DMU with the network structure. To the best of our knowledge, this study is the first attempt to detect and evaluate the congestion of the DMUs with the multi-function parallel network systems.

In this paper, the concept of congestion has been developed to deal with multi-function parallel network systems. For this purpose, firstly, the Production Possibility Set (PPS) has been developed corresponding to the multi-function parallel systems. Then, the concept of congestion is defined based on the developed PPS. In the following, the one-model linear programming problem has been proposed to detect the congestion of sub-units along with the overall congestion of DMUs. It has been proved that a DMU with a multi-function parallel network structure exhibits congestion as a whole if and only if there exists at least one subunit that exhibits congestion. Moreover, it has been shown that if a DMU exhibits congestion when considered as a black-box then, it also exhibits congestion according to the proposed model. Finally, the proposed model has been illustrated using a numerical example to examine the congestion of 8 hypothetical education institutions. Moreover, a real case study has been presented to investigate the congestion of 51 economic enterprises including 3 parallel subunits. In this case study, the results obtained from the proposed method have been compared with the results obtained from the traditional definition of congestion according to the black-box point of view.

## Author Contributions

**Data curation:** Bijan Rahmani Parchikolaei.

**Supervision:** Farhad Hosseinzadeh Lotfi, Mohsen Rostamy-Malkhalifeh.

**Validation:** Alireza Amirteimoori.

**Writing – original draft:** Sarvar Sadat Kassaei.

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
