## [Decision Letter · Decision Letter 0]

24 Apr 2023

PONE-D-22-34058Congestion in Multi-Function Parallel Network DEAPLOS ONE

Dear Dr. Lotfi,

Thank you for submitting your manuscript to PLOS ONE. After careful consideration, we feel that it has merit but does not fully meet PLOS ONE’s publication criteria as it currently stands. Therefore, we invite you to submit a revised version of the manuscript that addresses the points raised during the review process.

We look forward to receiving your revised manuscript.

Kind regards,

Thanh Ngo, Ph.D.

Academic Editor

PLOS ONE

Journal Requirements:

Reviewers' comments:

Reviewer's Responses to Questions

**Comments to the Author**

1. Is the manuscript technically sound, and do the data support the conclusions?

Reviewer #1: Yes

Reviewer #2: Yes

2. Has the statistical analysis been performed appropriately and rigorously? 

Reviewer #1: Yes

Reviewer #2: Yes

3. Have the authors made all data underlying the findings in their manuscript fully available?

Reviewer #1: Yes

Reviewer #2: Yes

4. Is the manuscript presented in an intelligible fashion and written in standard English?

Reviewer #1: Yes

Reviewer #2: Yes

5. Review Comments to the Author

Reviewer #1: The article entitled “Congestion in multi-function parallel network DEA”, was very well written. I recommend accepting it, however, I leave some recommendations to be observed, only in the section - Introduction.

In the last paragraph of the introduction, the numbering of the sections has been written in the correct sequence. However, the next section should appear as “2. Preliminaries” and not with the numbering “1. Preliminaries”.

Therefore, the numbering of ALL sections and subsections must be readjusted.

Reviewer #2: The author insight the congestion status of DMUs is one of the most significant topics in the DEA literature. I consider the problem of congestion interesting before performing the DEA analysis, however it is convenient that you mention a little the part of the tools used to run the proposed model, and present the data used since in the tablets you can only see figures and output boxes.

The DEA is mentioned in the literature review, but with the problem of congestion it is recommended to assess the technical inadequacy, but in this case the Stochastic Frontier Analysis SFA is more recommended (see suggested reference), I suggest that the author make an assessment at the regard. line 17-32

I suggest the following bibliography. So I would like that you add more references may 40 or 45.

[1] Zuniga Gonzalez CA and Jaramillo-Villanueva JL. Frontier model of the environmental inefficiency effects on livestock bioeconomy [version 1; peer review: awaiting peer review]. F1000Research 2022, 11:1382 (https://doi.org/10.12688/f1000research.128071.1)

6. PLOS authors have the option to publish the peer review history of their article (what does this mean?). If published, this will include your full peer review and any attached files.

Reviewer #1: No

Reviewer #2: No

---

## [Author Response · Author response to Decision Letter 0]

16 May 2023

PONE-D-22-34058

Title: Congestion in Multi-Function Parallel Network DEA

Dear Dr. Thanh Ngo,

Thank you so much for your time and efforts. We would like to sincerely appreciate the reviewers for carefully reviewing the manuscript and for their insightful comments and suggestions to improve the quality of the manuscript. We have incorporated these suggestions in the revised version and have addressed all the issues pointed out in the review report. All the mentioned comments along with their associated responses are described below. 

Finally, please note that all the changes/revisions made in the revised manuscript have been highlighted.

Yours sincerely,

Farhad Hosseinzadeh Lotfi

Response to the comments of Reviewer #1

(Comments) The article entitled “Congestion in multi-function parallel network DEA”, was very well written. I recommend accepting it, however, I leave some recommendations to be observed, only in the section - Introduction.

In the last paragraph of the introduction, the numbering of the sections has been written in the correct sequence. However, the next section should appear as “2. Preliminaries” and not with the numbering “1. Preliminaries”.

Therefore, the numbering of ALL sections and subsections must be readjusted.

The reviewer is absolutely right. We appreciate his/her comment for pointing it out. The numbering of all sections and subsections has been readjusted in the revised section. Now, the first section appears as “1. Introduction” and other sections have been numbered after it from Number 2. All the changed numbers are highlighted in the revised version.

Response to the comments of Reviewer #2

(Comments) The author insight the congestion status of DMUs is one of the most significant topics in the DEA literature. I consider the problem of congestion interesting before performing the DEA analysis, however it is convenient that you mention a little the part of the tools used to run the proposed model, and present the data used since in the tablets you can only see figures and output boxes.

We really appreciate the reviewer's efforts to provide valuable comments for improving the scientific level of the manuscript. About this comment, it should be noted that all the proposed models are linear programming problems that can be easily solved with existing software such as GAMS or MATLAB. However, according to the opinion of the respected reviewer and to clarify the matter for the readers, a part entitled "Remark 3.1" has been added to the manuscript on Page 9. In addition, two proposed models in the numerical example were discussed in more detail, which can be seen highlighted on Pages 15 and 16. If there is still any shortcoming, we will be grateful if you can guide us to correct it.

(Comments) The DEA is mentioned in the literature review, but with the problem of congestion it is recommended to assess the technical inadequacy, but in this case the Stochastic Frontier Analysis SFA is more recommended (see suggested reference), I suggest that the author make an assessment at the regard. line 17-32 I suggest the following bibliography. So I would like that you add more references may 40 or 45.

[1] Zuniga Gonzalez CA and Jaramillo-Villanueva JL. Frontier model of the environmental inefficiency effects on livestock bioeconomy [version 1; peer review: awaiting peer review]. F1000Research 2022, 11:1382 (https://doi.org/10.12688/f1000research.128071.1)

Again, we are very grateful for the constructive suggestion of the reviewer. We agree with his/her opinion. Accordingly, we dedicated the first paragraph of the manuscript to introduce the SFA and its applications. In this paragraph it has been mentioned that SFA is utilized to analyze the technical inefficiency in the framework of production functions, and, its main advantages are its capacity to accommodate statistical noise, such as measurement error, and its parametric specification of the technology, allowing standard statistical tests to be used. Moreover, regarding this issue, 5 new references (i.e. References [26], [27], [35], [41], and [42]) have been added to the manuscript by using the suggested reference. In the following, it has been mentioned that SFA is sensitive to a priori assumptions and requires a pre-specification of the functional form; then, for this reason, the non-parametric Data Envelopment Analysis (DEA) method can be used to measure the efficiency of homogenous Decision-Making Units (DMUs) without needing any specification of the functional form of the production function. All added parts have been highlighted.

---

## [Editor Report · Decision Letter 1]

26 May 2023

Congestion in Multi-Function Parallel Network DEA

PONE-D-22-34058R1

Dear Dr. Lotfi,

We’re pleased to inform you that your manuscript has been judged scientifically suitable for publication and will be formally accepted for publication once it meets all outstanding technical requirements.

Kind regards,

Thanh Ngo, Ph.D.

Academic Editor

PLOS ONE

---

## [Editor Report · Acceptance letter]

30 May 2023

PONE-D-22-34058R1 

Congestion in Multi-Function Parallel Network DEA 

Dear Dr. Lotfi:

I'm pleased to inform you that your manuscript has been deemed suitable for publication in PLOS ONE. Congratulations! Your manuscript is now with our production department. 

Kind regards, 

on behalf of

Dr. Thanh Ngo 

Academic Editor

PLOS ONE